# Mechanistic insights into Alpha-Synuclein binding to P2RX7: A molecular dynamic and docking study

**Mukesh Kumar** ⓘ*, **Kanchan Singh, Jayant Joshi, Shreya Sharma, Amit Kumar, Karuna Irungbam, Manish Mahawar, Mohini Saini**

ICAR-Indian Veterinary Research Institute, Bareilly, Uttar Pradesh, India

* mukesh.kumar6@icar.gov.in

## Abstract

Alpha-synucleinopathies, characterized by extracellular alpha-synuclein (αSyn or SNCA) accumulation and aggregation, have been linked to neurological disorders including Parkinson's disease and multiple system atrophy. P2RX7 is a non-selective cationic transmembrane purinergic receptor activated by elevated levels of extracellular ATP, which typically occurs during inflammatory conditions. Activation of P2RX7 by αSyn is implicated in neuronal degeneration, potentially causing pore dilation and increased inflammation. By integrating the data curation, molecular docking, and molecular dynamics (MD) simulations, along with structural analyses, we attempted to elucidate the molecular mechanisms and binding sites for P2RX7-αSyn interaction. We elucidated interactions between P2RX7 and the N-terminal domain (NTD) of αSyn. Utilizing cryo-EM structures of P2RX7 in ATP-bound and unbound states, we assessed αSyn's effect on P2RX7 structural and functional dynamics. Initially, the analyses revealed that αSyn interactomes are mainly involved in mitochondrial functions, while P2RX7 interactors are linked to receptor internalization and calcium transport. Molecular docking with six tools identified that αSyn-NTD fragments preferentially bind to the proximal region of P2RX7's transmembrane domain. Microsecond all atom MD simulations in a POPS lipid bilayer showed significant atomic fluctuations, particularly in the head region, lower body, and large loop of P2RX7's cytoplasmic domain. Secondary structure analysis indicated unfolding in regions related to pore dilation and receptor desensitization. Further by contact-based and solvent accessibility analyses, along with protein structure network (PSN) studies, we identified crucial residues involved in αSyn-P2RX7 interactions. This understanding enhances the knowledge of how αSyn and P2RX7 interactions take place, potentially contributing to neurodegenerative diseases, and could be instrumental in developing future preventive and therapeutic approaches.

## Introduction

The purinergic receptor P2RX7 and alpha-synuclein (αSyn or SNCA) are mainly expressed in neural tissues, individually or in lethal combination which causes many pathological consequences such as Parkinson's Disease (PD), neurodegeneration, dementia and brain tumours

**Data availability statement:** All the raw data, scripts, and main outputs of this study are available from figshare repository (https://doi.org/10.6084/m9.figshare.28152182).

**Funding:** This research is supported by the ICAR-Indian Veterinary Research Institute (https://www.ivri.nic.in/) under the funded project titled "Elucidating the role of alpha synuclein (SNCA) and cholesterol on regulating the P2RX7 pore dilation" [Project No. F.7-40/Biochem./2022-23/JD(R)]. The funders had no role in study design, data collection and analysis, decision to publish, or preparation of the manuscript.

**Competing interests:** No competing interest.

[1–3]. αSyn is susceptible to extracellular accumulations and aggregations due to gene over-expression, mutations such as (A30P, A53T), structural transitions [4–5] following structural and functional alterations, which can result in membrane damage and a pro-inflammatory response [2,6,7]. On the other hand, increased levels of extracellular ATP (eATP) which may occurs due to membrane damages, hyper-activate P2RX7 in brain cells, allowing for pore dilatation and intracellular accumulation of multivalent cations that ultimately result in necrosis and pro-inflammatory response [8,9]. Though these proteins exhibit distinct pathogenic pathways, they eventually result in an inflammatory response and cellular damage [10]. αSyn is an intrinsically disordered protein (IDP) that interacts with a variety of partners, including membrane proteins and lipids, to support physiological responses such as synaptic transmission, neuronal differentiation, growth, and dopamine synthesis [2,3]. One of these partners is P2RX7, which has recently been shown to interact directly with α-synuclein, leading to its activation and the subsequent accumulation of dysfunctional mitochondria in neural cells [11,12]. These dysfunctional mitochondria could be the source for reactive oxygen species (ROS) and pro-inflammatory responses that contribute to neuronal degeneration [13]. Though P2RX7 and αSyn have been linked to P2RX7 activation and pore dilatation, their molecular interactions and conformational changes and the way they interact in regulating pore dynamics remain unknown.

Studying the interactions between P2RX7 and αSyn is precisely challenging because, in addition to the complexity of the P2RX7 trans membrane protein and the highly disordered nature of αSyn, time scale limitations of atomic fluctuations and molecular interactions dealing with the dynamical changes pose great hindrance in unsheathing the meaningful information from conventional wet lab molecular biology study. However, it is now possible to predict protein-protein interactions with greater accuracy thanks to recent developments in proteomics and structural biology, improved computer efficiency, and more precise algorithms for structure predictions and free energy calculations [14–17]. About 100 experimental 3D structures of the αSyn alone or complexes with ligands, either in native or fibril forms, are available in the protein data bank (PDB) which could be a greater resource for capturing the atomistic views of structural and functional alteration associated to the PD and other neurodegenerative diseases [18]. Among the seven members of the purinergic family, P2RX7 is distinct because it has the longest cytoplasmic tail and has been suggested to be instrumental in pore dilatation and receptor desensitization [19,20]. The full-length rat P2RX7's ATP-bound and unbound cryoEM structure has been recently reported by McCarthy et al. 2019 and revealed many structural elements such as pore forming units, cysteine rich region (CRR), cytoplasmic cap and cytoplasmic pore together with structure of a unique cytoplasmic ballast [21]. This study also divulged the unprecedented role of palmitoylated juxta-transmembrane intracellular CRRs and demonstrated the stability of cytoplasmic cap regulates the receptor desensitization [21]. The cytoplasmic tail previously shown to be involved in protein-protein, protein-nucleotides, protein-lipids interactions regulating P2RX7 activations, receptor desensitization and pore dilations [22]. Nevertheless, still the structural and functional dynamics of this domain as well as the impact of dynamical alterations over the structural and functional attributes of P2RX7 has not been thoroughly addressed in the past due to the lack of a full-length experimental structure for this receptor.

Recent advances in computational efficiency and structure analysis algorithms, combined with the availability of well annotated full-length experimental structures of P2RX7 and αSyn, have enabled an atomic-level understanding of the dynamics of molecular interactions. This understanding is crucial for identifying the molecular determinants of these interactions.

In our current study, we have addressed the questions cited above using the state of art techniques including Molecular Docking, MD simulations and Structure Analysis Framework that we employed in our previous studies [23,24]. Initially, we curated and annotated the

αSyn and P2RX7 protein interactome and examined the pattern for binding sites preferred by their protein interactors. Later ATP bound (PDB ID 6U9W) and free forms (PDB ID 6U9V) of ratP2RX7 3D structures were employed as a template in our current work to construct the human P2RX7 model. This model was then further used in molecular docking and all atom MD simulation studies in conjunction with protein structure network analysis to better understand the conformational dynamics and identify the molecular determinants of P2RX7 and αSyn interactions. Our molecular docking and MD simulation analysis shows αSyn does have more propensity for binding to the transmembrane domain of P2RX7. Using multiple, distance-based analysis; we mapped the binding interfaces and narrowed down the residues for more in depth analysis for their binary interactions. We identified the critical residues which are involved in the hydrophobic interactions and salt bridges between P2RX7 and αSyn which could be pre-eminent not only for interactions but also in perpetuating the stability.

## Materials and methods

All the raw data, raw scripts and raw main outputs of this study are freely available in the figshare (10.6084/m9.figshare.28152182), GitHub (https://github.com/MukDAN/IVRI_P2RX7_SNCA_DATA) and Zenodo (Zenodo. https://doi.org/10.1101/2024.08.21.608916) repositories.

### Data curations and annotations of structure elements

The experimental 3D structures of ATP bound rat P2RX7 (open form; PDB ID: 6U9W) and ATP-Free rat P2RX7 (Closed form; PDB ID: 6U9V) are obtained from protein data banks (PDB) ([8,21]. Likewise, experimental 3D structure of human αSyn (PDB ID 1XQ8) was obtained from PDB [25]. Structural elements were annotated using the literature reviews [21,25–31].

### Data curation of αSyn and P2RX7 interactions and gene enrichment analysis

Both experimental as well as predicted protein interactors of αSyn and P2RX7 from five databases (IID, MINT, mentha, STRING and BioGrid) [32–36] as well as from literatures were curated and annotated for their interactions sites in αSyn and P2RX7 respectively. Total 1247 protein interactors of αSyn and 99 interactors of P2RX7 were used for gene enrichment analysis using enrichR webserver. Adjacent cut-off of > 0.05 was considered for analysis. Networks of the P2RX7 and αSyn interactomes were prepared by Cytoscape 3.1 software [37]. Structural binding sites were annotated from literatures and STRING databases.

### Molecular modelling

We utilized the MODELLER [38] program to obtain the trimerized human P2RX7 3D structures both open and closed form after reconstructing the missing residues and loops in cryoEM structures of rat P2RX7 (PDB ID 6U9W and 6U9V). The human P2RX7 protein sequence used during modelling was retrieved from Uniprot (ID: Q99572). The missing regions of rat P2RX7 were optimizing solely while leaving the rest of the structure intact. Selection of an appropriate model was based on Resprox resolution scores and the alignment of N-terminal domains toward intracellular sides. Models with finest resolution were selected for molecular docking with αSyn.

### Molecular docking

Both processed forms of human αSyn (PDB ID 1XQ8) and open and closed form of human P2RX7 models (hP2RX7-6U9W and hP2RX7-6U9V) were used for molecular docking using

five different docking tools; Galaxy-PepDock [39], Cluspro [40], pyDock [41], HPEPDOCK [42] and CABS-Dock [43]. Initially, we docked the small fragments (12,18 and 30 mer peptides) of 97 amino acid long N-terminal domain of αSyn over P2RX7 (6U9W). We examined the top 10 models from each docking tool using the Pymol visualization tool [44]. Finally, we docked αSyn over P2RX7 (hP2RX7-6U9W and hP2RX7-6U9V) using HDOCK web server [45] and the top most models with the highest scores was selected. The H-DOCK generated the N-terminal domain of αSyn complex with P2RX7 (Open; hP2RX7-6U9WSNCA and closed; hP2RX7-6U9V-SNCA) and remaining c-terminal disordered regions were discarded automatically.

## Modelling of trans membrane P2RX7-αSyn complex

The CHARMM GUI webserver [46] was used to model the final models of P2RX7-αSyn complexes (hP2RX7-6U9W-SNCA and hP2RX7-6U9V-SNCA) in POPS lipid bilayer (188 + 188 POPS) with 24.95% cholesterol (63 + 62 CHL) in TIP3P solvent system [47,48]. After energy minimization and six step equilibrations, well equilibrated models were subjected for the molecular dynamic simulations.

**Molecular dynamic (MD) simulations.** We conducted all-atom classical microsecond MD simulations for each replicate of apo hP2RX7 (6U9W and 6U9V) and two replicates of each αSyn complexes form (hP2RX7-6U9W-SNCA1-2 and hP2RX7-6U9V-SNCA1-2) using the CHARMM 36m force field in a TIP3P water system [48,49]. Energy minimization employed the steepest descent algorithm, with bond lengths involving hydrogen atoms constrained by the LINCS algorithm during equilibration. Particle Mesh Ewald (PME) was used for long-range electrostatic interactions with a real-space cut-off of 12 Å, which was also applied to Van der Waals interactions. Initial steps in the NVT ensemble applied Berendsen temperature coupling, while subsequent steps in the NPT ensemble used both Berendsen temperature and pressure coupling. The final 200 ns MD simulation was conducted in the NPT system using Nose-Hoover temperature coupling and Parrinello-Rahman pressure coupling. A time step of 2 fs was set for all simulations, with data recorded every 100 ps. MD trajectory analysis (including RMSD, RMSF, radius of gyration, solvent accessibility, minimum distance and PCA) was done by Gromacs software suite [23,24,50].

## Conformational dynamics analysis

The dynamic changes in P2RX7 upon αSyn binding are quantified using Root Mean Square Deviation (RMSD), Root mean square fluctuation (RMSF), Radius of Gyration (Rg) and Principal Component Analysis (PCA). The covariance matrix of Cα atomic fluctuations was generated to estimate the eigenvalues followed by eigenvectors and projected them into 2D graph to visualize the conformational overlap. Thereafter, we selected the PC1 for extracting the principal motions from the MD trajectories. The 175 ns and 75 ns concatenated trajectories of P2RX7 from both its apo and complexes forms were employed for PCA, allowing comparison within the same essential subspace using the GROMACS functions gmx covar and gmx anaeig [23,24,47,51].

**Secondary structure analysis.** Quantification of overall secondary structural elements was done by GROMACS embedded DSSP program [52] and PLUMED plugin [53,54] posteriori. Later, streamlined secondary structure contents were evaluated using the AlphaRMSD and ALPHABETA Collective variable (CV) available in PLUMED plugin posteriori [53].

**Contact analysis and protein structure network analysis (PSN).** Intermolecular and intramolecular contact analysis with cut-off 5Å was done by GROMACS gmx_mindist as well as gmx pair_dist and CONAN software suits [53,54]. PSN analysis from MD trajectory data of

P2RX7-αSyn complexes were done by PyInteraph and PyKnif2 tools [55] keeping the cut-off 5Å for Hydrophobic interactions, 4.5 Å for electrostatic interactions and 3.5Å for hydrogen bond interactions.

## Results

### Structural and Functional elements of αSyn and P2RX7

Detailed analysis, we described the 1) key structural and functional elements in P2RX7 that have been reported [21,26] to play a pivotal role in ATP binding and pore dilation, 2) αSyn structural elements that have been suggested to be involved in interactions with proteins and membranes. αSyn is divided into two domains; N-terminal domain (NTD: 1-98 aa) and highly polar C-terminal disordered domain (CTD: 99-140) [25] (S1A Fig). N-terminal domain is amphipathic and cationic in nature which is armed with imperfect repeats of KTKEGV residues and can establish electrostatic interactions with the negatively charged residues and membrane phospholipids. This domain consists of phospholipid (1-60), glycosphingolipids (34-45 aa) and cholesterol binding regions (high affinity cholesterol binding region: 67-78 aa and low affinity cholesterol binding regions: 32-43) and is the main element to interact with the membrane lipid rafts [56]. The end terminal of NTD contains highly hydrophobic region (67-94aa) known as "non-Aβ component of Alzheimer's disease amyloid plaque (NAC) which is culprit for multimerization and fibrillation of αSyn proteins [57]. Because of the amphipathic nature of NTD, we can visualize the hydrophobic and hydrophilic surface that is involved in molecular interactions. Concomitantly, P2RX7 is a dolphin shaped trimerized, double spanning trans membrane protein consisting of three structural domains: an external domain (ETD; 47-334 aa), two trans membrane domains (TMD1; 26–46 aa and TMD2; 335-355 aa) and an intracellular N-(1-25) and C-terminal (356-595) cytoplasmic domain [17]. ETD further segregates into head (111-169), upper body (70-92, 105-115 and 291-315), lower body (50-68, 94-106,188-209, 250-277 and 316-327), left flipper (278-292), right flipper (178-189, 235-250), and dorsal fin [27]. ETD contains highly conserved ATP binding sites (K64, K66, T189, N292, R294 and K311) which upon activation brings conformational changes, passes through lower body and the trans membrane domains, which is the main instrument for pore formation. Trans membrane domains, consisting TMD1 and TMD2 made up of α1 helix [24–48] and α6 helix [331–358]. TMD has a unique component known as the cytoplasmic cap (β_1, β0, and β15) formed by successive strands ($\beta_0$ and $\beta_{-1}$) in the N termini of two protomers and a strand ($\beta_{15}$) after TMD2 in the third protomer. The C-terminal cytoplasmic domain in P2RX7 is recently unravelled by CryoEM which is largest among all other purinergic proteins consisting of cysteine rich regions (CRR), cytoplasmic pore, cytoplasmic plug, large bulky cytoplasmic ballast and regions involved in interactions with lipids and GDP [21] (S1B Fig).

### P2RX7 and αSyn interactome and their interacting sites

To understand the molecular interactions between αSyn and P2RX7, we followed a combinatorial approach including data curation, molecular docking and MD simulations. In order to learn more about what part of alpha synuclein is generally involved in interactions with the proteins other than P2RX7, we first reviewed the interactions of Syn with 1247 protein interactors that have been curated from five databases mentha [32], BioGRID [33], STRING [34], IID [35] and MINT [36] and literatures via manual curations which previously followed in other work [23,24]. We took into account both interactors, whether they were supported by computational or experimental methods. However, interactors that had been demonstrated through experimentation received more attention and underwent in-depth research (S1A Table). In our curated datasets of αSy protein

interactomes, the gene enrichment analysis of by enrichR [58] R package was done using enrichment terms: GO-Biological process, GO-Cellular process, GO-Molecular function, Reactome 2022 and KEGG 202. We found that the top most GO terms belonged to the mitochondrial functions and homeostasis. Likewise, Reactome 2022 top 10 terms were mostly associated with immune response and infections whereas KEGG 2020 pathways were predominated by terms associated with Calcium ($Ca^{++}$) signalling and pathogenic infection. The detailed analysis is shown in the S2A Table. Furthermore, we wanted to determine which portion of the αSyn was typically involved in interactions with these binding partners. This was a crucial information for our current work since we needed to know how and which αSyn structural elements might interact with P2RX7. For this, we thoroughly investigated the experimentally determined αSyn interactors, their mode of binding, and the αSyn elements involved in the interactions. Hitherto, we have filtered out a total **107** no. of interactions which are annotated with C and N -terminal domain of αSyn, out of which **84** interactors interacted with C-terminal Domain and **24** with N-terminal domain. Our analysis demonstrates that the majority of the protein interactors are interacting with the CTD of αSyn compared to NTD as shown in Fig 1A. This is in line with what we anticipated since CTD is extremely disordered and richer in acidic amino acids. In parallel, we also curated the total 99 (S1B Table) P2RX7 interactors that are either experimentally confirmed or predicted. Gene enrichment analysis of P2RX7 interactors by enrichR [58] R package using GO-Bio, GO-Cellular, GO-Molecular function, Reactome 2022 and KEGG 2021 enrichment terms, we found that the top most term for Gene Ontology involves in Receptor Internalization and Bleb Assembly which are known for playing eminent role in membrane integrity as well as in homeostasis (S2B Table). The majority of P2RX7 are lacking binding site annotations, only few of them are annotated. Whatever is the case; most of them are found to be interacting with the cytoplasmic domain (Fig 1B).

## Alpha-synuclein (αSyn) interacts with the P2RX7 transmembrane domain (TMD)

After learning about the interactors and binding sites, we explored the interaction between extracellular α-synuclein and P2RX7 and the potential binding sites. Due to the fact that the external domain of the functional trimeric P2RX7 faces the extracellular environment, we speculated that the likelihood of ETD interacting with extracellular αSyn may be as high as possible. In light of these considerations, we aimed to reconstruct the human P2RX7 (hP2RX7) from available experimental structure of rat P2RX7 using MODELLER and perform a comprehensive blind molecular docking study involving both open and close formed of human P2RX7 models (hP2RX7-6U9W & hP2RX7-6U9V) and αSyn peptides. However, due to inherent limitations in the precision of force fields quantifying changes in binding free energy and constraints in sampling methods, the accuracy of docking diminishes significantly with increasing peptide length. To enhance the precision and reliability of our predictions, we adopted a two-pronged strategy. Firstly, we generated SNCA fragments of varying lengths (12 mer, 18 mer, 24 mer, and 30 mer) and conducted docking onto P2RX7 using five distinct docking tools.

Galaxy-PepDock [39], Cluspro [40], pyDock [41], HPEPDOCK [42] and CABS-Dock [43] to assess the pattern of preferred orientation of αSyn peptide fragments over P2RX7. Secondly, we finally docked the larger fragment of Syn (97aa) over hP2RX7 using H-DOCK [45], a tool that considers the hybrid algorithm of template-based modelling and *ab initio* blind docking (Fig 2A).

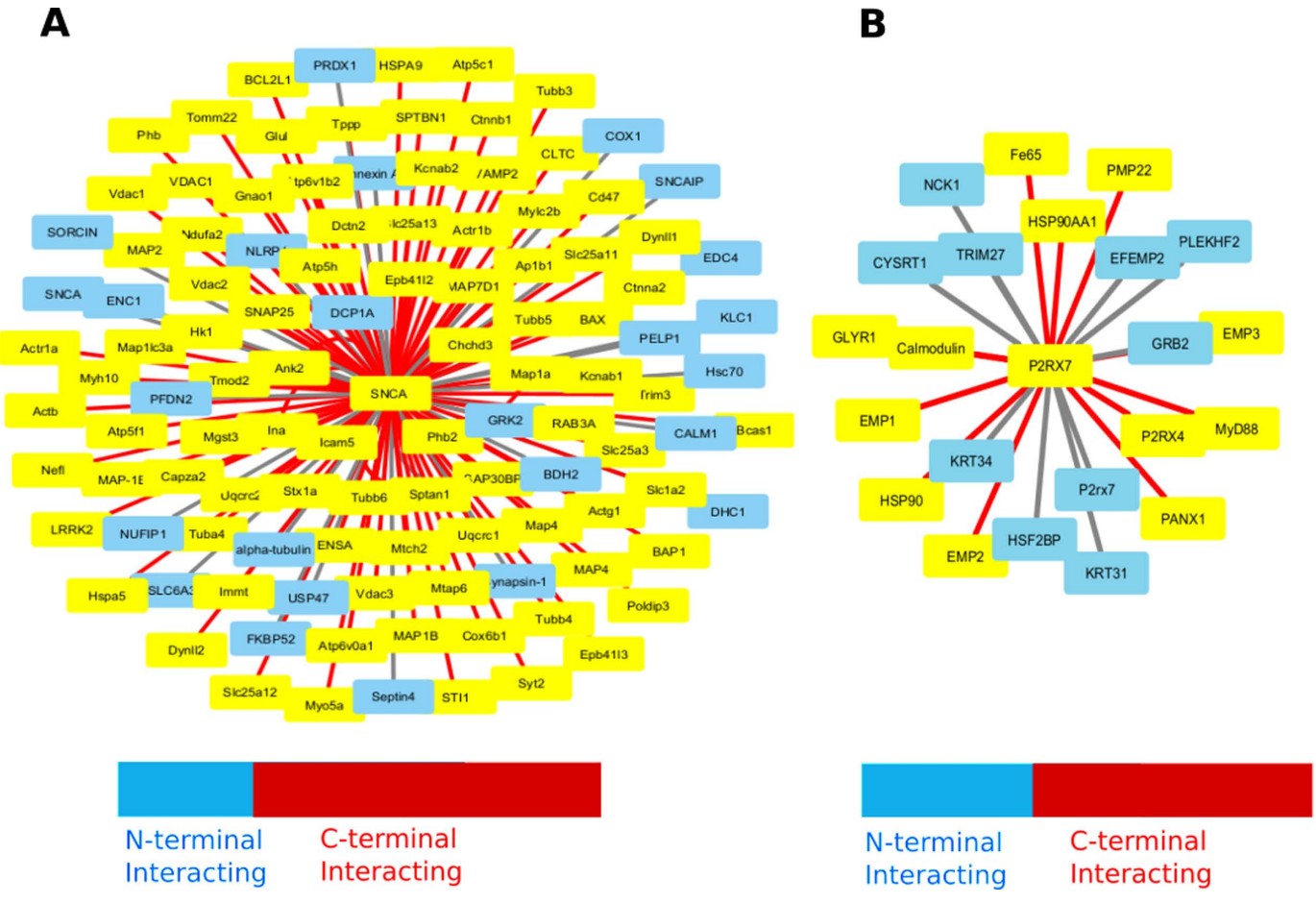

**Fig 1. αSyn and P2RX7 proteins interactome.** (A) Out of 1247 potential interacting proteins of αSyn (SNCA), 107 have been identified and annotated for their binding sites, either at the N-terminal or C-terminal domains of αSyn. (B) From 99 potential interacting proteins of P2RX7, 22 have been specifically annotated for their interactions with either the N-terminal or C-terminal domains of P2RX7. Proteins interacting with the N-terminal are highlighted in light blue, while those interacting with the C-terminal are marked in red.

This approach avoids relying on a single docking program and takes into account different algorithms that consider diverse structural and physicochemical attributes. Our findings revealed that a majority of the top models of these peptide fragments exhibits binding either to the region surrounding the transmembrane domains or to the interface between transmembrane domains. This observation is intriguing, especially considering the utilization of a variety of peptides with varying lengths. Following the experimentation with short peptide fragments using multiple docking tools, we proceeded to dock the larger fragment of the αSyn N-terminal domain (97 aa) (Fig 2B & Fig 2C). The first model which demonstrates the highest binding free energy changes with lower RMSD scores was selected for downstream analysis. This meticulous approach ensures a comprehensive examination of the interaction between αSyn and P2RX7, accounting for various peptide lengths and utilizing diverse docking tools to provide a thorough understanding. In our subsequent studies, we used the open (hP2RX7-6U9W-SNCA) and closed (hP2RX7-6U9V-SNCA) forms of the P2RX7 and αSyn complex models to obtain the microsecond molecular dynamics simulations in order to assess the stability of the models, identifying the molecular determinants of the interaction between P2RX7 and αSyn and quantifying the structural and functional dynamics changes exerted on P2RX7 upon αSyn interactions.

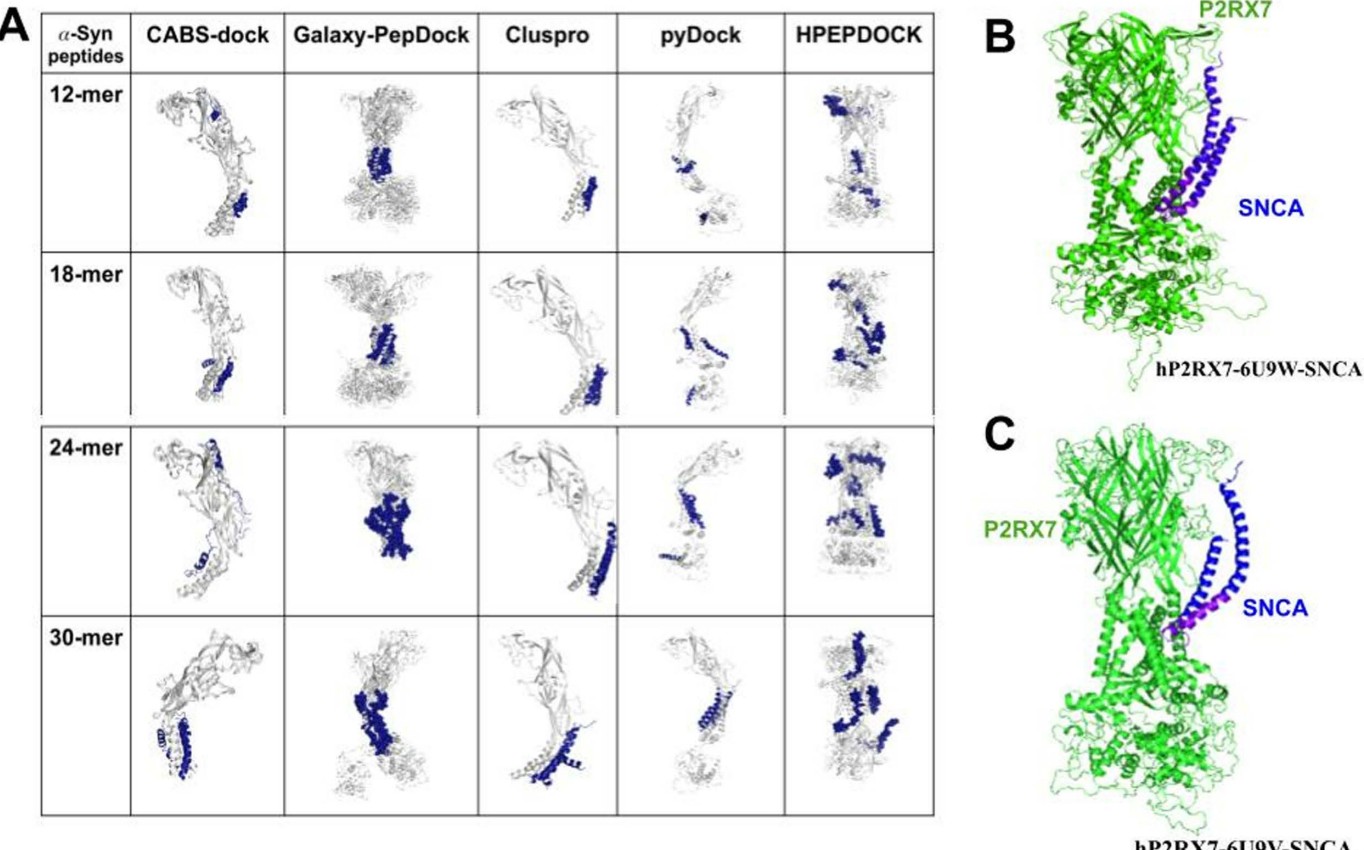

**Fig 2. αSyn peptides docking to P2RX7.** (A) top 10 model of four different fragments (12mer, 18mer, 24 mer and 30 mers peptides) of αSyn peptides docked over P2RX7 using five docking tools (Galaxy-PepDock, Cluspro, pyDock, Hpepdock and CABS-Dock**),** overall showing more propensity for transmembrane domains. (B & C) αSyn N-terminal domain containing 97 amino acids was docked over open (hP2RX7-6U9W) and closed forms (hP2RX7- 6U9V) of human P2RX7 model respectively using H-Dock webserver.

## Conformational dynamics of P2RX7-αSyn complex

The trans membrane protein P2RX7 establishes a complex with the intrinsically disordered αSyn protein. Similar to other complexes, the hP2RX7-αSyn complex undergoes periodic conformational changes under physiological conditions. These dynamic alterations, unveiling atomic fluctuations, as well as local and global conformational shifts in P2RX7 upon αSyn interaction, can be systematically quantified through molecular dynamics (MD) simulation [59,60]. The insights gained from these simulations contribute to defining the intricate structure-functional relationship within the molecular system. Before going into analysing the concatenated trajectories of apoP2RX7 and hP2RX7-αSyn complexes, we examined the evolutions of conformational space and atomic fluctuations of individual trajectories with the difference between their average structures using RMSD, RMSF, Rg and PCA metrics (S2A, S2B, S2C, S2D, S2E Figs). First evaluated the stability and evolution of the molecular system during simulation by calculating the backbone RMSD of αSyn alone, hP2RX7 alone and hP2RX7-αSyn complexes [61] in both open and closed forms of hP2RX7. The results indicate stable conformational evolution for both the apo hP2RX7 and hP2RX7-αSyn complexes across all models during the 200 ns simulation timeframe, as evidenced by stable RMSD profiles. However, the RMSD of αSyn exhibits higher values and greater fluctuations, which is expected

given its nature as an intrinsically disordered protein (S2A Fig). Thereafter, we evaluated the alterations in compactness of the molecular system during MD simulation by estimating the Rg. In our analysis, we found that compactness of αSyn is much poorer than hP2RX7 alone and hP2RX7- αSyn complexes (S2B Fig). However, it is comparatively more stable than RMSD during MD simulation. Other things, we noticed that compactness of P2RX7- αSyn is slightly lower than the hP2RX7 alone, possibly due to presence of αSyn. We further truncated the MD trajectory from 25ns to 200 ns and estimated the backbone RMSD and observe that the evolution of the molecular systems is quite stable in both close and open forms of apo-P2RX7 and their αSyn complexes except the first replicate of hP2RX7-SNCA-6U9V-I, which exhibited a larger average RMSD (~6.97 Å) until stabilizing after 125 ns, aligning with other models. We also observed the averages RMSD of hP2RX7αSyn complexes are relatively higher than the apoP2RX7 (Fig 3A). The comparatively higher RMSD observed may be attributed to the intrinsically disordered nature of *αSyn*, as we have seen large variation in RMSD of *αSyn* (S2A Fig). Next, we evaluated the per residual flexibilities during MD simulation by RMSF [62]. RMSF analysis, capturing residual flexibilities, consistently identified specific P2RX7 residues with higher flexibility, ranging from 4 to 20 Å across all models. We could not see the significant differences in residual flexibilities between apo P2RX7 and P2RX7-αSyn complexes. However, in all cases, flexible residues are predominantly confined in coil regions atop the external domain, encompassing the head and ATP binding sites (76-84, 122-137, 150-150), as well as in the extreme cytoplasmic ballast (447-458, 460-471, 585-595) (S2C Fig). The flexibility observed in these P2RX7 residues suggest their potential involvement in various inter- and intra-atomic contacts with neighbouring residues, a phenomenon that has been investigated in our subsequent experiments.

Thereafter, we analysed the conformational dynamics, global and local conformational changes of P2RX7 upon *αSyn* interactions by Principal Component Analysis (PCA) [63]. PCA utilizes collective coordinates to derive a lower-dimensional subspace where functionally relevant principal motions are likely to occur [51]. The subsequent PCA thus focused exclusively on conformational dynamic alterations in P2RX7 upon *αSyn* binding, with the exclusion of *αSyn* dynamics analysis as intrinsic limitations of classical MD simulation would not be enough to capture the relevant information associated with large variations. Prior to conducting an in-depth analysis of the truncated trajectories, we performed a preliminary examination of the spatio-temporal evolution of individual MD trajectories for both apo and P2RX7-SNCA complexes. Utilizing PCA on the Cα atoms, we derived eigenvalues and eigenvectors, subsequently projecting them onto two-dimensional graphs as PC1 vs. PC2 and PC1 vs. PC3 (S2F Fig). This analysis revealed that, over the course of the MD simulations, the conformational spaces of all complexes evolved effectively, encompassing a substantial range of time points. The projections indicated that both closed and open forms of apo P2RX7 occupied distinct and expansive conformational clusters, signifying notable conformational transitions. Conversely, the P2RX7-SNCA complexes demonstrated a progressive yet less diverse evolution, forming fewer clusters and undergoing more constrained conformational transitions. This suggests that the binding of SNCA may limit the dynamic range of P2RX7.Considering the stability observed in RMSD for all models from 25 ns onward, with the exception of model hP2RX7-6U9V-SNCA-I, which stabilized after 125 ns, we chose to further implement PCA on the covariance matrix of Cα atomic fluctuations on two molecular dynamics (MD) trajectories: one covering 25 to 200 ns and another shorter trajectory from 125 to 200 ns.

Initially, we examined 175 ns molecular dynamics (MD) trajectories of P2RX7 from both the apo and complex forms. Our analysis focused on the first three principal components, which collectively represented over 50% of the variation (S2D Fig). Shorter MD trajectories of the apo and complexes forms revealed variations of approximately 50% and 45%, respectively

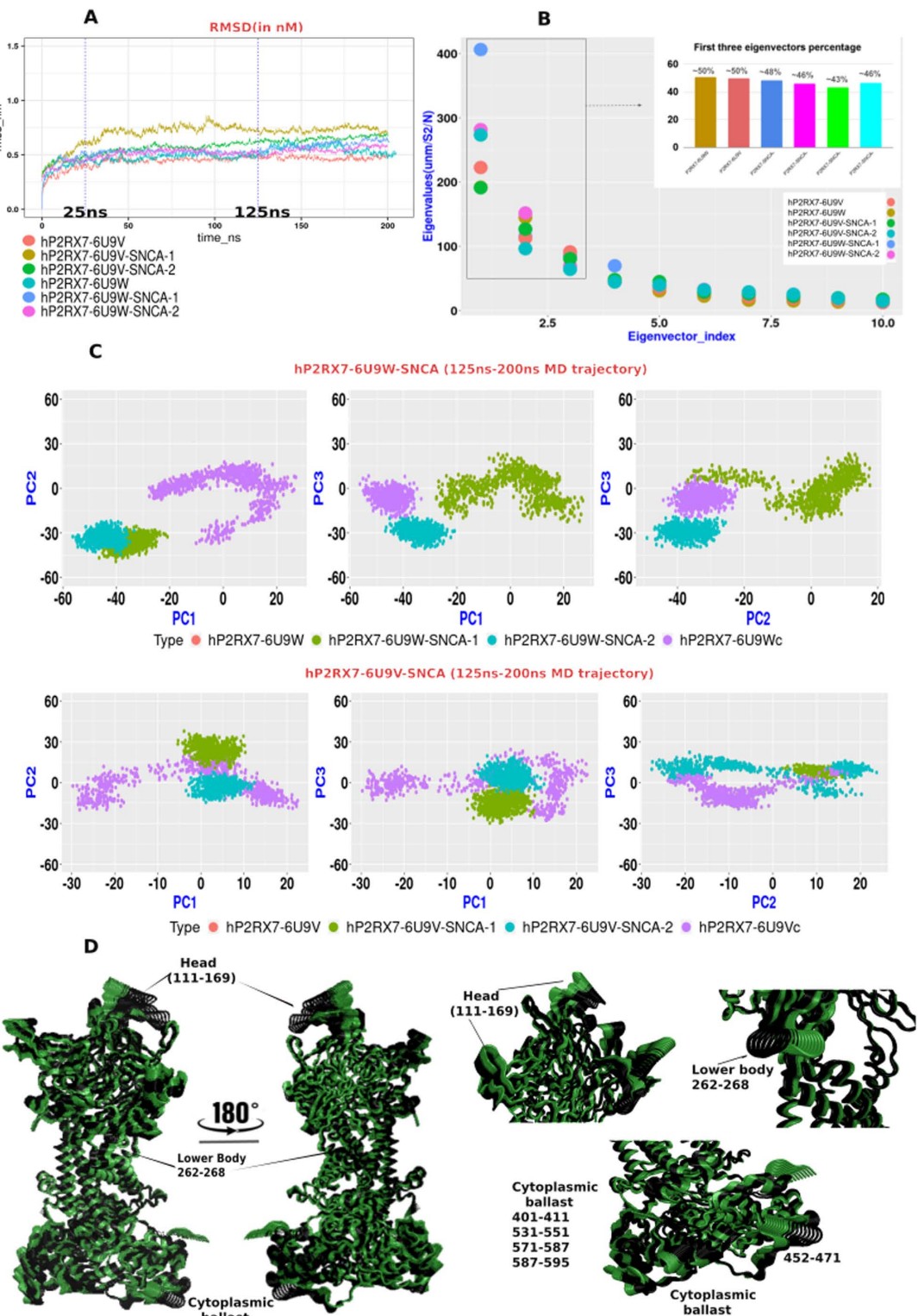

**Fig 3. Conformational dynamics of apoP2RX7 and P2RX7- SNCA complex.** (A) RMSD analysis of apoP2RX7 and P2RX7-SNCA complex models following 200 ns MD simulations shows that all models remain stable, except for hP2RX7-6U9V-SNCA-1, which exhibits a relatively higher RMSD but stabilizes from 125 ns onwards until 200 ns. The graph shows the distances for the closed and open forms of apoP2RX7 in dark orange (P2RX7-6U9V) and cornflower blue (P2RX7-6U9W),

while the corresponding SNCA complexes are shown in dark khaki (hP2RX7-SNCA-6U9V1), dark olive green (hP2RX7-SNCA-6U9V2), cornflower blue (hP2RX7-6U9W-SNCA-1), and magenta (hP2RX7-6U9W-SNCA-2). (B) Distribution of the first 10 eigenvectors from PCA analysis of a shorter MD trajectory shows the proportion of the first three eigenvectors for each model within the box, displayed in a bar diagram labelled with eigenvector percentages. The same colour codes of legends shown in the previous Figure. (C) PCA analysis was performed using concatenated trajectories of the respective apoP2RX7 and P2RX7-SNCA complexes and projected the PCs in 2D graph. The graph shows in upper panel, the open forms of apoP2RX7 in dark orange and purple (P2RX7-6U9W) while the corresponding SNCA complexes are shown in dark olive green (hP2RX7-SNCA-6U9W1), cornflower blue (hP2RX7-6U9W-SNCA-2). The same color code was depicted in lower panel for apo forms of closed P2RX7 and corresponding complexes. (D) A comparison of the principal motion extracted from PC1 of the open forms of apoP2RX7 (Green) and the corresponding P2RX7-SNCA complex (Black) reveals that the concerted motions in the head, lower body, and cytoplasmic ballast differ significantly from those in the apo form, indicating that SNCA binding induces distinct concerted motions.

(Fig 3B). Remarkably, even after reducing the concatenated trajectory length by 62.5%, the collective variances remained preserved. These findings suggest that essential collective motions associated with αSyn interactions are retained. Notably, the first three principal components captured significant conformational variance in both the longer and shorter trajectories, which are likely pertinent to P2RX7 functions. Given this, our next step involves exploring the conformational overlap among PC1, PC2, and PC3 to elucidate the dominant motions in different regions of P2RX7 upon αSyn interactions. In the longer MD trajectories, we observed partial overlap between the P2RX7-αSyn complexes and a subset of the apoP2RX7 near the central axis. However, the PC1 vs. PC2 plot of the hP2RX7-6U9V-SNCA complex showed noticeable deviations, indicating structural differences despite similarities in certain regions (S2E Fig). This suggests that while collective motions in the complexes also occur in the apoproteins, they are less pronounced. In contrast, shorter MD trajectories revealed distinct conformational differences compared to apoP2RX7, with complexes P2RX7 showing isolated pockets of conformations (Fig 3C). This suggests that αSyn interaction triggers unique protein motions, leading to discernible structural changes absent in the apo state. Comparing the principal motions from the first eigenvectors between open forms of complexes and apoP2RX7, we observed distinct coordinated motions in the Head region, Lower body, and large loop of the cytoplasmic domain mostly belonging to the chain B (Fig 3D). However, we didn't notice any notable changes in the closed state of the hP2RX7-αSyn complex, which aligns with what we've seen in previous comparisons using PC1 vs PC2 projections. These shared directional tendencies indicate concerted structural changes induced by αSyn binding, reflecting a functional response of P2RX7. The shorter MD trajectories capture these unique dynamics and pronounced αSyn influences, leading to observable structural alterations not seen in the apo state.

## P2RX7 pore dynamics upon αSyn binding

P2RX7 contains trans membrane domain (TMD) and cytoplasmic pores, both of which play an important role in transportation of extracellular cations. In the closed form, residues Q332, V335, S339, and S342 face the lumen of the trimerized P2RX7, where TMD2 forms a 0.1 Å radius pore. Upon ATP binding, the pore dilates to a minimum radius of 2.5 Å, allowing hydrated Na+ to pass through [21]. Likewise, P2RX7's c-terminal tail contains a distinct element known as the cytoplasmic ballast, comprising the α9–α16 alpha helices. The α9 helix, also referred to as the cytoplasmic plug, can fit into the approximately 14 Å wide cytoplasmic hole formed by α12 and α13. The outer boundary of the cytoplasmic pore is formed by α14 and α16, with α15 positioned at the top of the pore. In this structure D335 and D438 are important residues that regulate cytoplasmic plug occupation over cytoplasmic holes. Due to

the lack of high-quality, complete experimental structures of P2RX7, the dynamics of cytoplasmic and transmembrane pore formation, in both open and closed states, and its complex with SNCA, remain unstudied.

Previous studies used distance and radius of gyration (Rg) analyses of L351 in P2RX4 to monitor pore dilation upon ATP binding [49]. However, L346 in P2RX7, which occupies a similar position, is oriented outwardly toward TMD1, making it unsuitable for this analysis. Instead, we focused on residues S339 and S342, recently identified by McCarthy et al. (2019), which form the TMD pore's boundaries [21]. We focused in a) TMD pore dynamics followed by b) Cytoplasmic pore dynamics upon SNCA interaction by quantifying changes in pore diameters through measuring the varying distances and summing all the distances between critical residues with their corresponding chains involved in pore opening. Similarly, the Rg of residues S339 and S342 offers insights into dynamic changes during MD simulations. In both pairwise distance and Rg analyses we find the pore size is quite distinct throughout the simulations between open and closed forms of apoP2RX7. However, we could not observe any significant differences between the closed and open forms of the complexes except a slight reduction in the radius of gyration (Rg) at S339 and S342 in open forms of the P2RX7-SNCA complexes relative to apoP2RX7 (Fig 4A-C).

Similar trends were observed in the radius of gyration (Rg) analysis of S339-S342 peptide fragments, as shown in Fig 4D. We then investigated structural transitions within the cytoplasmic pore and plug, particularly focusing on the α9 helix, which acts as a cytoplasmic plug containing the critical residues D435 and D438. Changes in flexibility and distances of these residues may provide insights into cytoplasmic pore dynamics. To assess these alterations, we conducted pairwise distance analysis and Rg measurements between the residues of corresponding chains. The analysis revealed an increase in the distances between these residues after 150 ns in the open forms of apoP2RX7 and hP2RX7-6U9W-SNCA-2, with the exception of the hP2RX7-6U9W-SNCA-1 replicate. These findings are consistent with the flexibility measurements observed in the Rg analysis (S3A-D Fig). Both experiments align with the trends reported by McCarthy et al., 2019, where open forms exhibited greater dilation of the cytoplasmic pore and increased separation of the α9 helix from corresponding chains compared to the closed forms.

## Dynamical changes in the secondary structure of P2RX7 structural elements

To understand the dynamics of secondary structural (SS) elements in P2RX7 upon αSyn interactions, the secondary structure contents of the ETD, CTD and TMD were observed using the DSSP programme [52], followed by alphaRMSD of posteriori MD trajectory analysis by PLUMED plugin(5ss3). We first investigated the dynamics of secondary structure (SS) changes in the entire P2RX7 receptor between its open and closed forms in apoP2RX7 to distinguish the structural transitions of both conformations and to determine if ATP-mediated conformations are stable. Following this, we analysed the corresponding complexes with αSyn to identify whether the interaction with αSyn induces specific structural changes in P2RX7. At first, we observed, between hP2RX7-6U9W and hP2RX7-6U9V as the former depicted more affinity towards the turns whereas the latter was more inclined towards the B-sheets (S4 Fig). Furthermore, the CRR (~360-377aa) which existed as α-helix in the closed state in hP2RX7-6U9V, unfolded in the open state of hP2RX7-6U9W. Apart from that, the residues lying in the dorsal fin region (206234, 801-829, 1396-1424), were found to be more in the coiled state in the hP2RX7-6U9W which is different from the hP2RX7-6U9V dorsal fin residues preferred to reside in the turns. Moreover, the pertinent region of hP2RX7 in both the open and closed

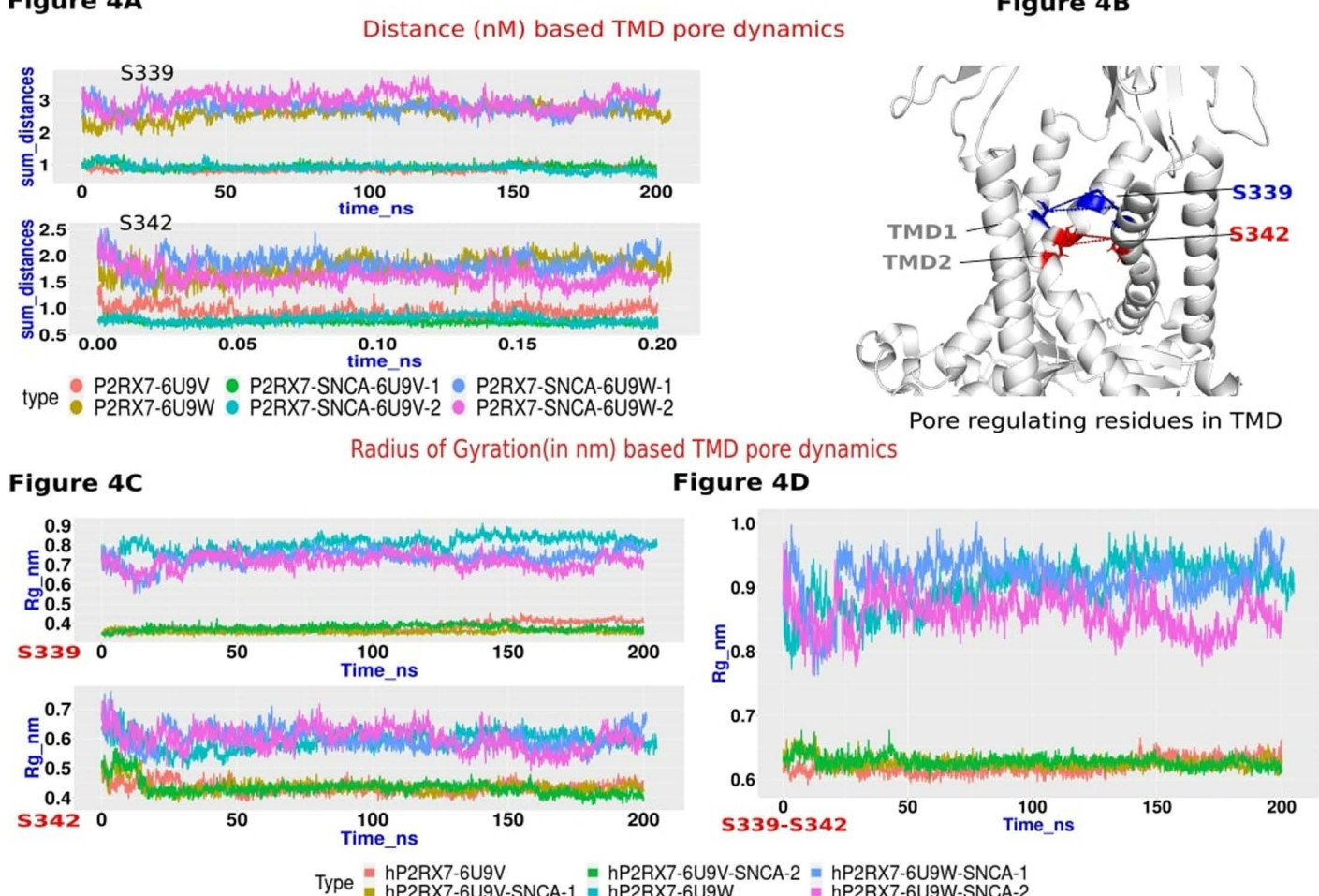

**Fig 4. TMD pore dynamics.** (A) This panel displays a pairwise distance analysis of the S339 and S342 residues across corresponding chains. The analysis was performed using the GROMACS function gmx pairdist, and the resulting distances were summed up over time and represented in a line graph. The graph shows the distances for the closed and open forms of apoP2RX7 in dark orange (P2RX7-6U9V) and dark khaki (P2RX7-6U9W), while the corresponding SNCA complexes are shown in aquamarine (hP2RX7-SNCA-6U9V1), dark olive green (hP2RX7-SNCA-6U9V2), cornflower blue (hP2RX7-6U9W-SNCA-1), and magenta (hP2RX76U9W-SNCA-2). (B) A cartoon diagram illustrates the TMD of P2RX7, highlighting crucial residues. Blue lines indicate the distances between the S339 residues of corresponding chains, while red lines show the distances between S342 residues across the chains. (C) The radius of gyration (Rg) analysis of residues S339 and S342 in chains A-C was performed using the GROMACS function gmx_gyrate and plotted as a line graph. The closed and open forms of apoP2RX7 are represented in dark orange (P2RX7-6U9V) and aquamarine (P2RX7-6U9W), respectively, while the SNCA complexes are shown in dark khaki (hP2RX7-SNCA-6U9V1), dark olive green (hP2RX7-SNCA-6U9V2), cornflower blue (hP2RX7-6U9W-SNCA-1), and magenta (hP2RX7-6U9W-SNCA-2). (D) The Rg analysis of TMD2 fragments containing the S339-S342 residues is shown, with the same color coding as in (C).

states is the cytoplasmic ballast where the residues in the open state unfolded a bit more than the closed state.

Next, we compared the apoP2RX7 with respective P2RX7-αSyn complexes and observed that in hP2RX7-6U9W-αSyn-I, the lower body region has more coil and 3-helix structures compared to hP2RX7-6U9V-αSyn I, which tends to have more β-sheet structures. Significantly, a change is observed in the Zn-binding region which is again an important hP2RX7 region whose functions have not been so far revealed. The bends and coils were more common in the hP2RX7-6U9V-αSyn -I than hP2RX7-6U9W-αSyn complex. Concomitantly, the TMD2 region is also different in the two complexes. Between the time scale of 110 to 150 ns

the number of coils in the complex I is found to be prominent as compared to the complex II. The TMD1 domain too showed presence of B-sheets in the hP2RX7-6U9V-αSyn I than hP2RX76U9W-αSyn I. Another interesting aspect is the CRR change as the latter depicted more number of 3-Helix. Cytoplasmic Ballast residues were also found to be different as it showed the presence of turns in the complex consisting of closed forms of P2RX7 and αSyn. Then, the pattern of hP2RX7-6U9V-SNCA II and hP2RX7-6U9W-SNCA II complex was almost identical with the hP2RX7-6U9W-SNCA I and hP2RX7-6U9V-SNCA I with slight differences in the CRR for the time span of 40ns to 200 ns where numbers of turns were found to be prominent.

Trans membrane domains are the pore forming units, alteration in the secondary structures contents and its dynamics directly impact the pore dilation. We noticed that while **α**1 helix (TMD1) appears stable in all models of P2RX7-αSyn complexes, **α**6 (TMD2) undergoes through large variations in the alpha helix contents in hP2RX7-6U9V-SNCA models

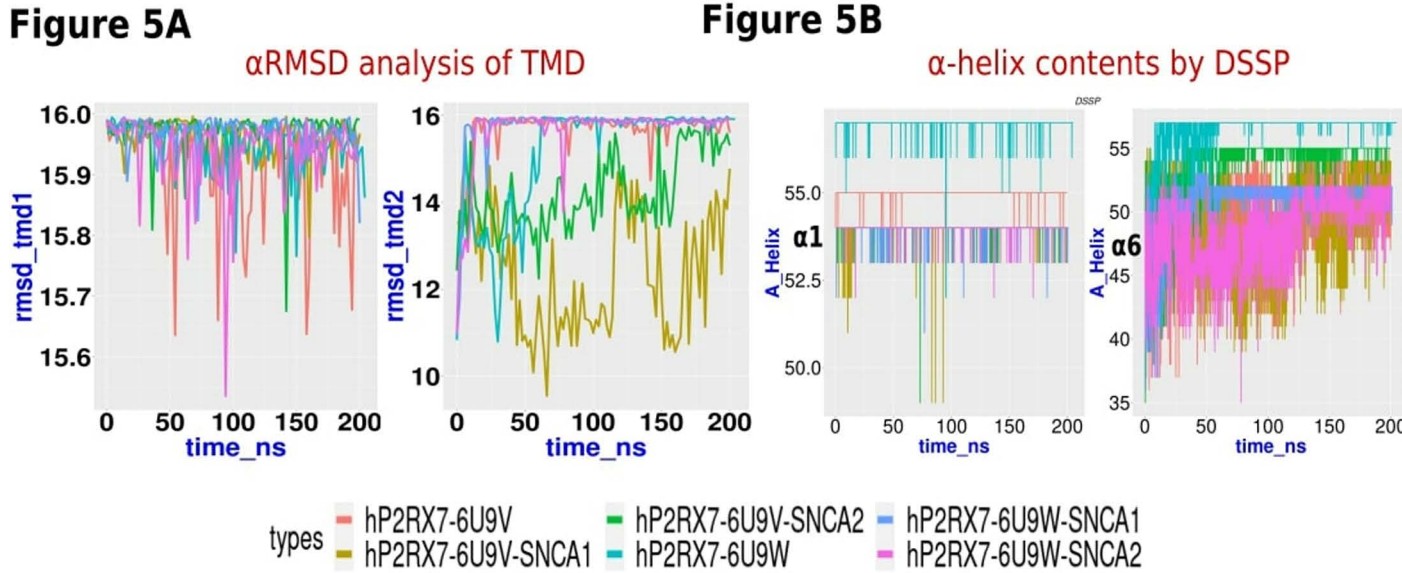

**Fig 5. Secondary structures contents analysis.** (A) The alphaRMSD analysis of the α1 and α6 helices in P2RX7's TMD1 and TMD2, respectively, depicted in the line plot, indicates that TMD1 remains relatively stable, while TMD2 exhibits significant variability. In particular, the closed P2RX7-αSyn (SNCA) complexes show a marked reduction in the α-helix content of TMD2. (B) The α-helix content analysis of both domains using the DSSP program supports the trends observed with the alphaRMSD collective variable from the PLUMED plugin. The consistent trends confirm the reduction in TMD2 α-helix content, especially in the closed P2RX7-αSyn complexes. The graphs are color-coded as follows: dark orange (P2RX7-6U9V) and aquamarine (P2RX7-6U9W) represents the closed and open forms of apoP2RX7, while dark khaki (hP2RX7-6U9V—SNCA-1), dark olive green (hP2RX7-6U9V- SNCA-2), cornflower blue (hP2RX76U9W-SNCA-1), and magenta (hP2RX7-6U9W-SNCA-2) represent the corresponding SNCA complexes. The cytoplasmic cap's stability is crucial for P2RX7 pore opening, as its unfolding may close the pore. ALPHABETA with phi and psi angle is a collective variable (CV) that can be used in metadynamic study for biased sampling. However, the same CV can also be used for quantifying the contents of alpha helix and beta strands using different reference values for standard phi and psi angle. Using ALPHABETA collective variables of PLUMED, we quantified the phi and psi angles of cytoplasmic cap beta strands (β-1, β0, β15) and found no significant differences between apoP2RX7 and hP2RX7-αSyn complexes (S5A-B Fig). P2RX7's c-terminal tail contains a distinct element known as the cytoplasmic ballast, comprising the α9–α16 alpha helices. The α9 helix, also referred to as the cytoplasmic plug, can fit into the approximately 14 Å wide cytoplasmic hole formed by α12 and α13. The outer boundary of the cytoplasmic pore is formed by α14 and α16, with α15 positioned at the top of the pore. The dynamical changes in the cytoplasmic pore and plug have not been extensively studied but are potentially very relevant to our research. Therefore, we quantified their structural transitions. In our analysis, we observed that alphaRMSD contents of α9, however we could not observe significant differences neither in apoP2RX7 forms nor in P2RX7-αSyn complexes (S5C Fig). Likewise we could not see the significant variations in alpha helical contents of α12 and α13 in both hP2RX7-6U9W-αSyn and hP2RX7-6U9V-αSyn. Cysteine rich region comprising the α8 and α7 helices are important sites for palmitoylation preventing the receptor desensitization. Our analysis shows CRR seems quite unstable in both models and could not notice significant differences.

compared to apo P2RX7 (Fig 5A). In parallel, we estimated the average α1 and **α**6 alteration during MD simulation using the DSSP program and observed the similar pattern found in PLUMED plugin analysis (Fig 5B).

### Identification of critical residues involved between P2RX7 and αSyn interactions

Interface residues are located between two proteins, facilitating their interaction and serving as the foundation for distance-based identification methods. Initially exposed to solvent, these residues become buried upon protein-protein interactions, potentially underpinning Solvent Accessible Surface Area (SASA) methods [64]. Initially, we examined the residues involved in the interaction between these two proteins upon docking using minimum distance analysis with 5Å cut-off between P2RX7 and αSyn. Later evaluated the stability of these interacting residues during MD simulation by measuring the distances and non-covalent interactions. Monitoring the stability of the docked αSyn-P2RX7 complex is crucial, as docked peptides can sometimes be unstable and escape the binding sites. To achieve this, we utilized microsecond all-atom molecular dynamics (MD) simulations coupled with distance-based analysis using Gromacs software [50], contact analysis via CONAN [54], and protein structure network (PSN) analysis through PyInteraph2 and PyInKnife2 tools [55]. The Gromacs based minimum distance analysis method considers any atoms within the specified distance between the two groups as constituting contact. Our analysis revealed that both molecules remained stable across all complexes, maintaining close proximity without any peptides escaping the P2RX7 binding pocket (S6A Fig). We identified 47 residues in hP2RX7-6U9W-SNCA model and 23 residues in the hP2RX7-6U9V-SNCA model exist within a 5Å cut-off. However, after molecular dynamics (MD) simulations, the number of residues involved in interactions significantly increased. Specifically, 390 residues in the open form and 286 residues in the closed form of P2RX7, out of a total of 1785 residues, were found to be in the vicinity of αSyn. These residues were predominantly located near TMD1, TMD2, the dorsal fin, and parts of the lower body, as well as the left and right flippers on both chain A and chain B (S6B Fig). However, a notable limitation of the Gromacs-based minimum distance analysis is its inability to assess the stability and persistence of these interactions over time. This critical dynamic information—such as the duration and frequency of contact formations between hP2RX7 and αSyn—remains unquantified. Consequently, transient contacts may be overrepresented, potentially complicating the identification of the most significant interaction residues. To address these challenges and enhance the reliability of our findings, we employed contact analysis using the same distance cut-off with the CONAN program [54]. This tool calculates both the persistence and frequency of contact formations, offering a more comprehensive view of interactions. Heat maps of contact analysis of representative residue αSyn-A90 with P2RX7 residues are shown in S6C Fig. We found a total of 38 residues and 27 residues common in interfaces of hP2RX7-6U9W-αSyn and hP2RX7-6U9V-αSyn (Fig 6A) respectively, out of which 22 residues are common in both complexes. We found mostly the residues of TMD1 (31-35), TMD2 (336-337,340,344), some part of the dorsal fin (220-225) and lower body (261-263,266 and 270) of chain A, are in stable contact with αSyn residues. While many residues of chain B loosely come in contact with αSyn. Solvent accessible surface area (SASA) is vital for analysing interfacial residues, as it measures the protein surface area exposed to solvent, typically water.

This analysis determines whether residues are exposed or buried within the protein structure. Residues with high SASA values are more exposed to solvent, influencing their potential interactions with other molecules. In protein-protein interactions, such as between hP2RX7 and αSyn, SASA analysis is essential for understanding the involvement of interfacial residues in binding

**Fig 6. Molecular Determinants of P2RX7 and SNCA Interactions.** In panels (A and B), the P2RX7 residues that are within 5Å of α-Syn, identified through contact analysis, were assessed for solvent accessibility using gmx-sasa from MD trajectories. The results for both hP2RX7-6U9W-SNCA and hP2RX7-6U9V-SNCA complexes are depicted, showing solvent exposure levels of the residues. Highly solvent-exposed residues are represented as dark blue spheres with a sasa value greater than 0.6, moderately accessible residues are shown in light blue with sasa values between 0.3 and 0.6, and buried residues are indicated by red spheres with sasa values

less than 0.3. Panels (C and D) illustrate the intermolecular interactions between P2RX7 (green) and α-Syn (red) in the hP2RX7-6U9W-SNCA and hP2RX7-6U9V-SNCA complexes. The interactions, which include hydrophobic interactions, salt bridges, and hydrogen bonds, are represented as sticks: hydrogen bonds are shown in sky blue, salt bridges in orange and hydrophobic interactions in blue.

and interaction processes. To assess this, we used gmx sasa to calculate solvent accessibility for P2RX7 residues. We found that 80 residues in hP2RX7-6U9W-αSyn and 89 residues in hP2RX7-6U9V-αSyn have high solvent accessibility (>0.3). These residues were further categorized based on their solvent accessibility: buried (<0.3), accessible (0.3-0.6), and highly accessible (>0.6). (Fig 6A and B). Thereafter, specifically, we focused on residues common to both hP2RX7-6U9W-αSyn and hP2RX7-6U9V-αSyn complexes, relevant to αSyn interaction (S6D Fig).

After streamlining the P2RX7 and αSyn residues using contact and solvent accessibility analysis, we further decided to examine the protein structure network properties chiefly hub residues and connected components as well as type of intermolecular interaction (such as hydrophobic, hydrogen bond and salt bridge interactions) between these residues using PyInteraph2 and PyInKnife2 tools that could be relevant to their stability and molecular functions. PSN analysis is an important tool to analyse the stability and conformational dynamics of the proteins where amino acid residues are considered as nodes and connection between them as edges [65]. First, we examined the hub residues and connected components (also known as clusters) in the psn generated using a centre of mass-based contact between the neighbouring residues keeping 5 Å cut-off, which could be involved in maintaining the stability of the P2RX7-αSyn complex. Hub residues are considered which connect either to three or greater than three neighbouring amino acid residues whereas clusters are the strongly connected set of residues. Someone can also consider clusters as graphs within the graph. In our analysis, we observed that hub residues and connected components are not significantly altered upon αSyn interaction with the open and closed form of P2RX7 (S7A- B Fig). However, we found that P2RX7 residues A176, C499 in open form and C477, C488, C499, C573 and C506 in closed form are the hub residues carrying largest degree (k >= 6) whereas hub residues (A11, K43 and A89) in the αSyn (k >= 4) were examined. These residues could be involved in maintaining the stability of the P2RX7-αSyn complex. We also examined intermolecular interactions such as the salt-bridge, hydrogen bond and hydrophobic intermolecular interactions within 4.5Å, 3.5Å and 5Å respective distance cutoff after excluding all the interaction that exists below 20% persistence (Table 1). Closed forms of P2RX7 form more hydrogen-bond interaction with αSyn residues where predominantly R264, D329 and D48 were found to be interacting with E28 and K43 respectively. Only one residue in the open state of P2RX7 showed affinity towards the hydrogen bond interaction which is Y40 with H50 of αSyn. Similar pattern was observed for the salt bridge interactions where the closed form of P2RX7 prominently interacts with the αSyn residues which are R264, D329, D48, K220, and E237 with the E28, K43, E57 and K60 respectively (Fig 6C & Fig 6D). Furthermore, the highest interaction seen between the two were the hydrophobic interactions. It is observed that prominently the residues in the closed form of P2RX7 F328, F266, L45, I208, P210, V26, L209, and L333 interacted with A29, A18, V26, V70, A69, V46, I88, and A29 with the αSyn. However, the open form of P2RX7 with residues V35, I330, P210, L209, I37, L212, V285, L262 and V37 also showed good degree of interaction within the αSyn residues A29, A56, A30, A78, V52/55, V49, V82, V55, V63, V10, V77 and A53.

## Discussion

P2RX7, akin to a "sleeping dragon" has potent cytotoxic effects primarily in neural tissues and macrophages. It regulates synaptic transmission, neural remodelling, and ATP-mediated

**Table 1.  Non-covalent interactions between P2RX7 and αSyn.**

| S.No. | P2RX7 | SNCA | persistence (>=20%) | Type of Interactions | Replicates |
|---|---|---|---|---|---|
| 1 | ARG-264 | GLU-28 | 56.62169 | Hydrogen_bond_AD | hP2RX7-6U9V-SNCA1 |
| 2 | ARG-264 | GLU-28 | 60.11994 | Hydrogen_bond_AD | hP2RX7-6U9V-SNCA1 |
| 3 | ASP-329 | LYS-43 | 21.43928 | Hydrogen_bond_AD | hP2RX7-6U9V-SNCA1 |
| 4 | ASP-329 | LYS-43 | 20.13993 | Hydrogen_bond_AD | hP2RX7-6U9V-SNCA1 |
| 5 | TYR-40 | HSD-50 | 35.455 | Hydrogen_bond_BD | hP2RX7-6U9W-SNCA1 |
| 6 | ASP-48 | LYS-43 | 32.33383 | Hydrogen_bond_BD | hP2RX7-6U9V-SNCA1 |
| 7 | ASP-48 | LYS-43 | 20.83958 | Hydrogen_bond_BD | hP2RX7-6U9V-SNCA1 |
| 8 | ASP-263 | LYS-10 | 57.13575 | Salt_brige_AD | hP2RX7-6U9W-SNCA1 |
| 9 | ASP-263 | LYS-21 | 23.62009 | Salt_brige_AD | hP2RX7-6U9W-SNCA1 |
| 10 | ARG-264 | GLU-28 | 92.15392 | Salt_brige_AD | hP2RX7-6U9V-SNCA1 |
| 11 | ASP-329 | LYS-43 | 65.96702 | Salt_brige_AD | hP2RX7-6U9V-SNCA1 |
| 12 | ARG-264 | GLU-28 | 32.33383 | Salt_brige_AD | hP2RX7-6U9V-SNCA2 |
| 13 | ASP-48 | LYS-43 | 69.81509 | Salt_brige_BD | hP2RX7-6U9V-SNCA1 |
| 14 | LYS-220 | GLU-57 | 22.58871 | Salt_brige_BD | hP2RX7-6U9V-SNCA1 |
| 15 | GLU-237 | LYS-60 | 27.78611 | Salt_brige_BD | hP2RX7-6U9V-SNCA1 |
| 16 | VAL-35 | ALA-29 | 97.50125 | Hydrophobic_contacts_AD | hP2RX7-6U9W-SNCA2 |
| 17 | PHE-328 | ALA-29 | 96.95152 | Hydrophobic_contacts_AD | hP2RX7-6U9V-SNCA2 |
| 18 | PHE-328 | ALA-29 | 95.65217 | Hydrophobic_contacts_AD | hP2RX7-6U9V-SNCA1 |
| 19 | PHE-266 | ALA-18 | 92.55372 | Hydrophobic_contacts_AD | hP2RX7-6U9V-SNCA1 |
| 20 | PHE-266 | ALA-18 | 76.76162 | Hydrophobic_contacts_AD | hP2RX7-6U9V-SNCA2 |
| 21 | VAL-35 | ALA-29 | 75.73347 | Hydrophobic_contacts_AD | hP2RX7-6U9W-SNCA1 |
| 22 | ILE-330 | ALA-56 | 72.70015 | Hydrophobic_contacts_AD | hP2RX7-6U9W-SNCA1 |
| 23 | LEU-45 | VAL-26 | 64.26787 | Hydrophobic_contacts_AD | hP2RX7-6U9V-SNCA2 |
| 24 | LEU-45 | VAL-26 | 62.96852 | Hydrophobic_contacts_AD | hP2RX7-6U9V-SNCA1 |
| 25 | VAL-35 | ALA-30 | 60.31825 | Hydrophobic_contacts_AD | hP2RX7-6U9W-SNCA1 |
| 26 | ILE-208 | VAL-70 | 57.32134 | Hydrophobic_contacts_AD | hP2RX7-6U9V-SNCA2 |
| 27 | PRO-210 | ALA-69 | 57.02149 | Hydrophobic_contacts_AD | hP2RX7-6U9V-SNCA2 |
| 28 | PRO-210 | ALA-78 | 56.02199 | Hydrophobic_contacts_AD | hP2RX7-6U9W-SNCA2 |
| 29 | LEU-209 | ALA-78 | 44.22789 | Hydrophobic_contacts_AD | hP2RX7-6U9W-SNCA2 |
| 30 | ILE-330 | VAL-52 | 42.67866 | Hydrophobic_contacts_AD | hP2RX7-6U9W-SNCA2 |
| 31 | ILE-330 | VAL-55 | 40.62969 | Hydrophobic_contacts_AD | hP2RX7-6U9W-SNCA2 |
| 32 | PRO-210 | ALA-78 | 39.5823 | Hydrophobic_contacts_AD | hP2RX7-6U9W-SNCA1 |
| 33 | VAL-46 | VAL-26 | 37.78111 | Hydrophobic_contacts_AD | hP2RX7-6U9V-SNCA1 |
| 34 | LEU-209 | ILE-88 | 35.43228 | Hydrophobic_contacts_AD | hP2RX7-6U9V-SNCA1 |
| 35 | VAL-46 | VAL-26 | 35.18241 | Hydrophobic_contacts_AD | hP2RX7-6U9V-SNCA2 |
| 36 | LEU-209 | ALA-78 | 34.90801 | Hydrophobic_contacts_AD | hP2RX7-6U9W-SNCA1 |
| 37 | ILE-37 | VAL-49 | 34.41074 | Hydrophobic_contacts_BD | hP2RX7-6U9W-SNCA1 |
| 38 | LEU-212 | VAL-82 | 31.12879 | Hydrophobic_contacts_AD | hP2RX7-6U9W-SNCA1 |
| 39 | ILE-330 | VAL-55 | 30.5818 | Hydrophobic_contacts_AD | hP2RX7-6U9W-SNCA1 |
| 40 | ILE-330 | ALA-56 | 30.38481 | Hydrophobic_contacts_AD | hP2RX7-6U9W-SNCA2 |
| 41 | VAL-285 | ALA-78 | 29.23538 | Hydrophobic_contacts_BD | hP2RX7-6U9W-SNCA2 |
| 42 | LEU-262 | VAL-63 | 28.73563 | Hydrophobic_contacts_AD | hP2RX7-6U9W-SNCA2 |
| 43 | VAL-10 | VAL-37 | 27.84684 | Hydrophobic_contacts_CD | hP2RX7-6U9W-SNCA1 |
| 44 | PRO-210 | VAL-70 | 27.08646 | Hydrophobic_contacts_AD | hP2RX7-6U9V-SNCA2 |
| 45 | VAL-35 | ALA-30 | 26.38681 | Hydrophobic_contacts_AD | hP2RX7-6U9W-SNCA2 |
| 46 | PRO-210 | VAL-77 | 24.03798 | Hydrophobic_contacts_AD | hP2RX7-6U9W-SNCA2 |
| 47 | ILE-330 | ALA-53 | 23.98801 | Hydrophobic_contacts_AD | hP2RX7-6U9W-SNCA2 |

*(Continued)*

**Table 1.** (Continued)

| S.No. | P2RX7 | SNCA | persistence (>=20%) | Type of Interactions | Replicates |
|---|---|---|---|---|---|
| 48 | LEU-333 | ALA-29 | 21.28936 | Hydrophobic_contacts_AD | hP2RX7-6U9V-SNCA1 |

lysis of antigen-presenting cells [8,66]. Similarly, the neuronal protein αSyn plays a key role in synaptic vesicle remodelling, trafficking, and neurotransmitter release. Extracellular accumulations of aggregate αSyn known for synucleinopathy as well as Parkinson's disease. The heightened activity of P2RX7 coupled with αSyn accumulation poses a threat to neural balance, contributing to detrimental outcomes such as microglial activation and subsequent dopaminergic neuron loss [1]. Wet-lab studies have shown that the interaction between P2RX7 and extracellular αSyn leads to mitochondrial dysfunction, oxidative stress, and inflammation [12]. However, the atomic-level details of these molecular interactions and their structural and functional implications remain unexplored.

Our research investigates the binding mechanisms and molecular changes in P2RX7 due to interactions with αSyn, highlighting their importance in neural pathogenesis. This work provides critical insights into these molecular interactions, deepening our understanding and potentially leading to new therapeutic strategies for diseases like Parkinson's, Alzheimer's, and certain cancers such as glioblastoma, glioma, pancreatic cancer, mammary tumours.

Our study began by exploring the structural and functional elements of P2RX7 and αSyn, followed by compiling a detailed list of proteins interacting with both, focusing on their binding sites and functional roles. Gene enrichment analysis of P2RX7 interactome revealed terms related to vital processes like receptor remodelling and inflammatory regulation, essential for neural receptor functions. Conversely, αSyn interactors were associated primarily with mitochondrial functions, crucial for ROS generation. Alterations in these interactors could affect mitochondrial function, P2RX7 and α-Syn stability, leading to oxidative stress, protein aggregation, and subsequent inflammation as previously shown in some studies [1,67,68].

The direct aggregation of αSyn due to P2RX7 interaction remains unconfirmed. However, studies suggest that the P2RX7-NLRP3 complex regulates αSyn levels in neural mononuclear cells of Parkinson's patients, while interactions between extracellular αSyn and P2RX7 may induce mitochondrial dysfunction, oxidative stress, and αSyn structural changes, promoting aggregation [12,67]. Additionally, extracellular αSyn facilitates P2RX7 and Pannexin-1 activation and recruitment respectively in neurons, promoting ATP release and stabilizing ATP levels by inhibiting extracellular Ecto-ATPase [11]. Similarly, another study shows interaction of extracellular αSyn not only facilitate the activation and recruitment of P2RX7 and Pannexin-1 respectably in neuronal cells and facilitating the ATP release but also stabilize the ATP level by inhibiting the extracellular Ecto-ATPase [11]. Overviewing the binding sites of these interactors shows unique patterns of C-terminal regions of both proteins, interacting with most of the protein interactors. Interestingly, the C-terminal regions are inherently more disordered in both proteins, which is common as many disordered regions contain short linear interaction motifs (SLiMs), critical elements for molecular interactions [69,70]. The C-terminal domain of P2RX7 features a CRR near TMD2 implicated in palmitoylation for receptor stability. Additionally, it contains a cytoplasmic ballast with 120 residues, assumed to play roles in receptor orientation, pore dilation and initiation of cytolytic signals [21,71]. This domain also houses a dinuclear zinc ion complex and a guanine dinucleotide phosphate (GDP) binding site [21] whose function is unknown.

After overviewing interaction pattern of interactome of both proteins, we explored extensively the molecular docking, employing various methods such as CABS-DOCK, Galaxy

Paddock, Cluspro, pyDock, H-Dock and hierarchical algorithms with MD simulations (HPEPDOCK), revealed that N-terminal domain of αSyn interactions predominantly occurs within the transmembrane domains of P2RX7, particularly in close proximity to the lower body and dorsal fins rich in hydrophobic residues. These findings align with observations regarding the structural and physicochemical properties of both αSyn and P2RX7. Specifically, the N-terminal domain (NTD) of αSyn exhibits amphipathic properties, with regions like the KTKEQ sequence and hydrophobic surfaces displaying affinity for the transmembrane domain (TMD) of P2RX7 which is enriched with the non-polar residues. Furthermore, the strong affinity of αSyn for membrane phospholipids and cholesterol suggests that its binding sites are likely located in amphipathic or hydrophobic-rich regions, such as the transmembrane domain or areas adjacent to it on P2RX7.

Our docking studies corroborate similar interaction patterns, supported by transient speculation from Zelentsova et al. (2022) in a recent review article [72]. Next, evaluated the stability of the hP2RX7-αSyn complex and their conformational dynamics using MD simulations in order to annotate the spatiotemporal evolution, molecular players of the interactions and functionally relevant dominant motions occurring [48,60] in P2RX7 upon αSyn interactions. The stability and convergence of the molecular system's evolution during MD simulation were assessed using RMSD to ensure accurate molecular property evaluations and to reduce errors and misleading interpretations. The RMSD analysis revealed, overall stability of all the apo and P2RX7- αSyn complexes except for hP2RX7-6U9V-SNCA-I, which initially had higher RMSD but stabilized later. The relatively higher RMSD in protein complexes compared to apoP2RX7 suggests that the presence of αSyn induces structural fluctuations, causing P2RX7 to adopt a more dynamic conformation and resulting in increased atomic positional variability over time (Fig 3A). Later, residual flexibilities were mapped by RMSF and found no significant differences between apo P2RX7 and hP2RX7-αSyn complexes. However, we could observe interesting uniform patterns in flexible residues of P2RX7 primarily clustered in confined coil regions atop the external domain and the extreme cytoplasmic region. Residual flexibilities indicate dynamic residue movements potentially involved in molecular interactions, with high RMSF values suggesting potential binding partners [73]. Although surveys in the ELM database [74] identify interacting motifs in flexible regions, their involvement in protein-protein interactions with αSyn remains elusive. In our previous study, we have successfully employed PCA for conformational dynamics to quantify the functionally relevant dominant motion in MD trajectories of LC3B and ULK1 protein and annotated their significance in structural and functional properties [23,24]. Similar methodology has also been employed here to delve into atomic fluctuations and dominant motions accurately. PCA was applied to two MD trajectories: 25-200 ns and 125-200 ns. This was prompted by a slight increase in P2RX7's RMSD during αSyn binding from 25 to 125 ns in one hP2RX7-6U9V-αSyn replicate. Both trajectories capture the significant proportion of the conformational variations required for annotating the structural features and principle motions. The collective variances remain preserved despite trajectory length reduction, indicating retention of essential collective motions associated with αSyn interactions. Plotting the PC1 vs PC2 shows partial overlap indicating the structural differences encountered upon αSyn interaction than apoP2RX7 despite similarities in certain regions. Comparing the principal motions between complexes and apoP2RX7, distinct coordinated motions were observed in the Head region, Lower body, and large loop of the cytoplasmic domain, mostly belonging to chain B. Principal motions occurring in these regions are crucial, as they are fundamental for agonist and antagonist binding, as well as for transmitting conformational changes from the head region to the transmembrane domain. The lower body plays a key role in conveying these changes and enabling radial expansion of the extracellular vestibule, leading to an iris-like dilation

of transmembrane helices upon ATP binding [75]. We probed the dynamical changes in the outer and inner boundaries of P2RX7-TMD by measuring the Rg and distances between key residues and their corresponding chain forming these boundaries which were not significantly altered between open and closed forms of P2RX7-αSyn complexes. We observed a slight increase in the compaction of the extracellular boundary involving S339 upon αSyn interaction compared to apoP2RX7. Despite this compaction, there is no significant alteration in the structural integrity or pore dynamics, as the sum distances between the S339 trimer remain unchanged. Moreover, the minor changes in compactness could not generate enough force to alter the pore diameters. We did not find comparable variations in the intracellular boundaries influenced by S342. This disparity might stem from S339's proximity to the disorder-prone lower body region of P2RX7, where even minor modifications could have substantial effects on neighbouring dynamics. In contrast, the S342 residue is centrally located within the transmembrane domain (TMD), closer to the stable cytoplasmic cap and distant from the extracellular disordered region. This positioning likely stabilizes S342, shielding it from significant atomic fluctuations caused by SNCA interactions. While we did not see substantial changes in pore dynamics, it is clear that the predominant protein motions take place in the external domain, particularly in the lower body, dorsal fins, and head regions during the interaction with SNCA. These motions could be significant for pore opening, as they are similar to the pathways for conformational induction associated with ATP-mediated pore dilation. Additional research is needed to investigate this further.

The absence of detected alterations could be attributed to the extended timescales required for these molecular events, necessitating longer observation periods. Alternatively, classical MD simulations may face challenges in detecting rare events due to inadequate sampling, especially those associated with high energy barriers or residing in global energy minima. Biased sampling techniques such as metadynamics or umbrella sampling could mitigate these limitations as demonstrated in our previous study [76].

Dynamical changes in secondary structures were quantified by using the DSSP and PLUMED plugin. The secondary structure analysis of pore forming units (α1 and α6 helices) shows unfolding of the TMD2- α6 in closed form of P2RX7 as compared to open form upon αSyn interactions. However, TMD1-α1 remains intact and stable in both forms. Interestingly, despite the unfolding of TMD2, pore dilation remained unaffected, reinforcing our previous assumption regarding limitations in conformational sampling. The other structural elements such as the cytoplasmic pore and cytoplasmic plugs were found stable despite being adjacent to the many disordered regions which were seen as quite flexible. CRR in all systems are found quite unstable and there are no conclusive differences observed.

To quantify the dynamics of intermolecular interactions, distance-based contact analysis and protein structure network (PSN) combined with solvent-accessible surface area (SASA) analysis were utilized. This approach identified critical residues of P2RX7 and SNCA that maintain persistent close contacts and are highly solvent-accessible, primarily mediating binding through hydrophobic interactions. Our PSN analysis revealed no significant changes in the topological properties, such as hub residues and clusters, between the open and closed forms of P2RX7-αSyn complexes. However, several hub residues were identified in both type of P2RX7-αSyn complexes. The closed form had relatively more hub residues than the open form, likely due to greater heterogeneity in its conformational space, as indicated by our conformational dynamics analysis. Furthermore, analyses revealed key residues involved in non-covalent interactions, with hydrophobic interactions serving as the foundation. Hydrogen bonds and salt bridges, including those with residues R264, D48, D263, D329 in P2RX7 and E28, K43, K10 in αSyn, enhance these interactions. Specifically, salt bridges between the anionic E28 and the cationic K43, K10 residues on αSyn strengthen binding, aligning

with findings from many researchers [77–79]. Most of these residues are confined in trans membrane domains, lower body and dorsal fins of the P2RX7 which are shown as the major pathways for the conformational signaling for P2RX7 activation and pore dilations [75]. These residues may serve as key molecular determinants facilitating the interactions between P2RX7 and αSyn. Such interactions have been previously identified as crucial for conferring conformational specificity and significantly contributing to molecular recognition and catalysis [80,81]. The observed structural complementarity provides a molecular basis for the P2RX7-αSyn interactions, similar to tendencies observed in our previous study [76].

Overall, our study provides a comprehensive view of the P2RX7 and αSyn interactomes, detailing their molecular functions and binding sites. We uncovered a novel interaction mechanism between P2RX7 and αSyn, identifying key molecular determinants and relevant structural and functional motions in P2RX7 induced by αSyn. Although we did not capture pore dynamics and some structural element dynamics—likely requiring longer MD simulations or enhanced sampling algorithms—these are under consideration. Future research may involve in silico saturation mutagenesis and wet lab experiments to further explore how changes in these determinants affect binding stability and affinity.

## Supporting information

**S1 Fig. Structural features annotation.** (A) SNCA (PDB ID 1XQ8) containing NTD (1-98aa) and CTD (99-140). The NTD possess Phospholipid binding region (1-60), Low affinity cholesterol binding region (32-43), Glycosphingolipid (GSL) binding region (34-45aa), high affinity cholesterol binding region (67-78) and hydrophobic core(61-95) comprising non-amyloid β-component (NAC). (B) P2RX7 (PDB ID 6U9W) comprises an external domain (ETD; 47-334 aa), two trans membrane domains (TMD1; 26-46 aa and TMD2; 335-355 aa) and an intracellular N- (1-25) and C-terminal (356-595) cytoplasmic domain (17). ETD further divided into head (111-169), upper body (70-92, 105-115 and 291-315), lower body (50-68, 94-106,188-209, 250-277 and 316-327), left flipper (278-292), right flipper (178-189, 235-250), and dorsal fin (206-234). ATP binding sites comprises K64, K66, T189, N292, R294 and K311 residues followed by subsidiary residues (R276, R277, K193, D280, Y288, L191, I214, I228, E186 and N187). The TMD consisting of TMD1 and TMD2 made up of α1 helix (24-48) and $\alpha_6$ helix (331-358) respectively. TMD is containing cytoplasmic cap ($\beta_{-1}$, $\beta_0$, and $\beta_{15}$). The Cytoplasmic N-terminal comprises 1-25 amino acids whereas C-terminal domain (358-595aa) comprises cysteine rich regions (CRRs), hexagonal ~ 14 Å widened cytoplasmic pore formed by $\alpha_{12}$, $\alpha_{13}$, cytoplasmic plug ($\alpha_9$), large bulky cytoplasmic ballast (α9–α16 and β16–β1) harbouring lipids and GDP binding sites.
(PDF)

**S2 Fig. Dynamical changes induced by αSyn on P2RX7 in larger trajectory.** (A) Backbone RMSD of αSyn alone, P2RX7 alone and P2RX7-αSyn complex of all the open and close forms of P2RX7 with and without αSyn were estimated for 200 ns time frame. The higher RMSD of αSyn alone is the distinguished in the line graph as compared to apoP2RX7 and P2RX7-SNCA complexes. The topmost left and right upper panel shows RMSD of closed (hP2RX7-6U9V) and open forms (hP2RX7-6U9W) of apoP2RX7 as orange line graph. The left and right middle panel shows RMSD of SNCA alone (blue), hP2RX7 alone (orange) and hP2RX7-SNCA complex (green) in line graph of respective closed and open forms of P2RX7-SNCA complex of replicate-1(hP2RX7-6U9V-SNCA1 & hP2RX7-6U9W-SNCA1). Similarly, the left and right bottom panel shows RMSD of SNCA alone (blue), hP2RX7 alone (orange) and hP2RX7-SNCA complex (green) in line graph of respective closed and open forms of P2RX7-SNCA complex of replicate-2 (hP2RX7-6U9V-SNCA2 & hP2RX7-6U9W-SNCA2). (B) The radius

of gyration (Rg) analysis reveals that αSyn alone is less compact compared to both apo P2RX7 and the P2RX7-SNCA complexes. Interestingly, apo P2RX7 exhibits a slightly higher Rg than the protein complexes, indicating a marginally more compact structure. The color scheme used aligns with that of the RMSD analysis. (C) RMSF of apoP2RX7 and P2RX7-SNCA complex models after 200 ns MD simulations. There are certain regions in P2RX7 in P2RX7 in both apo and complexes which are quite flexible. Most of these regions (Head, ATP binding sites, cytoplasmic ballast). The open form of P2RX7 (hP2RX7-6U9W) is depicted in aquamarine, while the closed form (hP2RX7-6U9V) is shown in dark orange. Their respective complexes are represented as follows: hP2RX7-6U9W-SNCA1 and hP2RX7-6U9W-SNCA2 in sky blue and pink, and hP2RX7-6U9V-SNCA1 and hP2RX7-6U9V-SNCA2 in khaki and green. (D) Distribution of first 10 eigenvectors of all the models after PCA analysis of 175 ns MD trajectories (25-200ns). First three eigenvectors of each model within the box show their proportion in the bar diagram labelled with eigenvector percentage analysis. The color schemes are same as followed in RMSF analysis. (E) Concatenated MD trajectory of respective apoP2RX7 and P2RX7-SNCA complexes of larger 175 ns MD simulations were used for PCA analysis. The projection of PC1 vs PC2, PC1 vs PC3, and PC2 vs PC3 for concatenated trajectories of apoP2RX7 and the P2RX7-αSyn (SNCA) complex using eigenvectors shows that certain subsets of conformations overlap in the apoP2RX7 and P2RX7-αSyn complexes, particularly in the central regions of the plots. This overlap suggests structural similarities between the apoP2RX7 and the P2RX7-αSyn complex. However, in the PC1 vs PC2 plot, isolated pockets are located away from the centre, indicating structural dissimilarities. This observation implies that while there are common structural features, some regions of the conformational space exhibit notable differences between the Apo and bound states. The colour codes are same as shown in Fig 3C. (F) Time and space evolution of conformational space of individual MD trajectories of open and close forms of P2RX7-SNCA complexes for 200ns MD simulations. The eigenvectors were obtained by PCA analysis and there after projected the PC1 vs PC2 and PC1 vs PC3 in 2D space. The upper left and right panel shows PC1 vs PC2 of close (hP2RX7-6U9V) and open forms (hP2RX7-6U9W) of P2RX7 with their corresponding SNCA complexes (hP2RX7-6U9W_SNCA1/2 & hP2RX7-6U9V-SNCA1/2). The grey colour scattered plot shows the PC1vs PC2 of apoP2RX7 and all the SNCA complexes. The colour transition from black to blue (apoP2RX7), black to red blue (P2RX7-SNCA-Complex-1), and black to green (P2RX7-SNCA-Complex-2), indicates time progression from 0 to 200 timescales. Likewise, PC1 vs PC3 of open and close forms were shown in the lower left and right panel and were depicted by the same colour pattern in relation to different timeframes. The results show that conformational space of all the complexes are evolved well during in different time points of MD simulation. Both PC1 vs PC2 and PC1 vs PC3 shows the conformational space of apo P2RX7 close and open forms are evolved well and exist in different clusters showing distinct conformational evolutions during MD simulations. Likewise, P2RX7-SNCA complexes were also evolved well but less diversified with fewer clusters and progressively but narrowly undergoes different conformational transitions. The results demonstrate that the conformational space of all complexes evolves effectively at different time points during the MD simulation. The PCA plots (PC1 vs. PC2 and PC1 vs. PC3) reveal that both the closed and open forms of apo P2RX7 exhibit well-defined evolution of conformational space, existing in distinct clusters that show divergent conformational transitions throughout the simulation. In contrast, the P2RX7-SNCA complexes also exhibit significant conformational evolution; however, they are less diversified, with fewer clusters observed. These complexes undergo progressive, yet more narrowly confined, conformational transitions compared to the apo form, suggesting a more constrained dynamic range due to the binding of SNCA.
(PDF)

**S3 Fig. Cytoplasmic pore dynamics.** (A) Perform a pairwise distance analysis for residues D435 and D438 in chain A, comparing these distances to the corresponding residues in chains B and C within both the apo and P2RX7-SNCA complexes using GROMACS software. After computing the distances, sum them separately for D435 and D438 and visualize the results with a line plot. (B) Illustrate the cytoplasmic pore structure, focusing on the α12- and α13-helices, and the cytoplasmic plug represented by the α9-helix, using cartoon diagrams. (C and D) Conduct radius of gyration (Rg) analysis for residues D435 and D438 in both the apo and P2RX7-SNCA complexes.
(PDF)

**S4 Fig. Assessment of secondary structural changes in HP2RX7 upon α interaction by DSSP during 200ns MD simulations.** The upper left and right panel shows secondary structural changes in open (hP2RX7-6U9W) and close (hP2RX7-6U9V) forms of apoP2RX7. The secondary structural changes in first replicate of hP2RX7-SNCA complexes of both open and close forms were shon in the middle left and right panel. Likewise, the bottom left and right panel shows second replicate of open and close forms of hP2RX7-SNCA complexes. In the figure, beta sheet, alpha helix, turn, 5-Helix, Bend and Beta Bridge are indicated by red, blue, yellow purple, green and dark grey colour whereas coil with no color.
(PDF)

**S5 Fig. The evaluation of cytoplasmic cap stability.** A) The Alpha-Beta Collective variables from the PLUMED plugin were used to assess the alpha and beta strand contents by analyzing standard phi and psi angles of -135° and 135° in both the apo and P2RX7-SNCA complexes. The results are displayed as a line plot. The graph shows in upper panel, the open forms of apoP2RX7 in dark orange while the corresponding SNCA complexes are shown in dark olive green hP2RX7- 6U9W- SNCA-1) and cornflower blue (hP2RX7- 6U9W- SNCA-2). The same colour code was depicted in lower panel for apo forms of closed P2RX7 and corresponding complexes (hP2RX7- 6U9V1- SNCA-1 and hP2RX7- 6U9V1- SNCA-2). B) Cartoon diagrams illustrate the cytoplasmic cap, which is composed of β-1, β0, and β15 beta strands, colour blue, maroon, and green, respectively. C) The α-helix content of the cytoplasmic pore (α12 and α13) and the cytoplasmic plug (α9) was calculated using αRMSD and presented as a line graph. The color code was same as shown in figure A.
(PDF)

**S6 Fig. The evaluation of contacts between P2RX7 and α-Syn.** A) The minimum distance analysis between α-Syn and P2RX7 showing quite close proximity. The results are displayed as a line plot. The graph shows corresponding P2RX7-SNCA complexes are shown in dark orange (hP2RX7- 6U9V- SNCA-1), dark olive green (hP2RX7- 6U9V- SNCA-2), corn blue (hP2RX7- 6U9W- SNCA-1) and purple (hP2RX7- 6U9W- SNCA-2). B) Minimum distances of each residues of P2RX7 to the α-Syn shows certain region of P2RX7 quite close to the α-Syn, most of these regions are trans membrane domains in chain A and chain B. The same colour codes of legend were followed shown in previous figure. C) The contact analysis by CONAN using 5Å distance cut-off shows persistence of contacts and number of contact formation. D) Solvent Accessibility Surface Area (SASA) analysis of P2RX7 residues shows proximity to the α-Syn. Most of the residues show higher solvent accessibility. The graph in left panel shows the open forms of apoP2RX7 in dark orange while the corresponding SNCA complexes are shown in dark olive green hP2RX7- 6U9W- SNCA-1) and cornflower blue (hP2RX7- 6U9W- SNCA-2). The same colour code was depicted in right panel for apo forms of closed P2RX7 and corresponding complexes (hP2RX7- 6U9V1- SNCA-1 and hP2RX7- 6U9V1- SNCA-2).

(PDF)

**S7 Fig. Hub and connected component analysis of P2RX7-SNCA Complex using Protein Structure Network (PSN) Analysis.** (A) The number of hub residues identified from the PSN analysis of P2RX7-SNCA complexes, specifically hP2RX7-6U9W-SNCA and hP2RX7-6U9V-SNCA. The PSN was constructed using contact information derived from salt bridge interactions (cut-off 4.5 Å), hydrogen bonds (cut-off 3.5 Å), and hydrophobic contacts (cut-off 5 Å) obtained from MD trajectories. These hub residues are crucial for maintaining the structural integrity and function of the protein complex. (B) The top 5 connected components identified from the PSN analysis of the hP2RX7-6U9W-SNCA and hP2RX7-6U9V-SNCA complexes. These components represent the most interconnected and stable regions of the protein structure during the 200 ns MD simulations, indicating key regions of structural and functional significance within the complex.
(PDF)

**S1A Table. SNCA interactome.**
(XLSX)

**S1B Table. P2RX7 interactome.**
(XLSX)

**S2A Table. SNCA Interactome gene enrichment analysis.**
(XLSX)

**S2B Table. P2RX7 Interactome gene enrichment analysis.**
(XLSX)

## Acknowledgments

We thank the Director, ICAR-IVRI and Joint Director Research ICAR-IVRI for their kind support. We thank the head, Division of Biochemistry for his valuable support and cooperation to execute the research project. We would like to thank the faculty of Biochemistry, Dr. Praveen Singh, Dr. Ajay Kumar and Dr. Meeta Saxena for their valuable guidance and useful feedback. We thanked technical officer Puja and attendant Riyasat Ji for his valuable support during conducting research in the lab. We would like to thank Dr. Himanshu Khandelia, Asso. Professor, South Denmark University, Denmark for his valuable suggestions during MD data analysis.

## Author contributions

**Conceptualization:** Mukesh Kumar.

**Data curation:** Mukesh Kumar, Kanchan Singh, Shreya Sharma.

**Formal analysis:** Mukesh Kumar, Shreya Sharma, Jayant Joshi.

**Funding acquisition:** Mukesh Kumar, Karuna Irungbam.

**Investigation:** Mukesh Kumar, Shreya Sharma, Jayant Joshi.

**Methodology:** Mukesh Kumar.

**Project administration:** Mukesh Kumar, Karuna Irungbam.

**Resources:** Amit Kumar.

**Software:** Mukesh Kumar.

**Supervision:** Mukesh Kumar, Amit Kumar, Manish Mahawar, Mohini Saini.

**Visualization:** Mukesh Kumar.

**Writing – original draft:** Mukesh Kumar, Kanchan Singh, Shreya Sharma, Karuna Irungbam, Manish Mahawar.

**Writing – review & editing:** Mukesh Kumar, Kanchan Singh, Shreya Sharma, Karuna Irungbam, Manish Mahawar, Mohini Saini.

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
