## [Decision Letter · Decision Letter 0]

12 Nov 2024

PONE-D-24-40709Insights from Molecular Docking and Dynamics Simulations of P2RX7- Syn ComplexPLOS ONE

Dear Dr. Kumar,

Thank you for submitting your manuscript to PLOS ONE. After careful consideration, we feel that it has merit but does not fully meet PLOS ONE’s publication criteria as it currently stands. Therefore, we invite you to submit a revised version of the manuscript that addresses the points raised during the review process.

We look forward to receiving your revised manuscript.

Kind regards,

Yusuf Oloruntoyin Ayipo, Ph.D

Academic Editor

PLOS ONE

“Mukesh Kumar

No.F.7-40/Biochem./2022-23/JD(R))

ICAR-Indian Veterinary Research Institute

https://www.ivri.nic.in/

No”

“Our present manuscript is supported by the institute funded project entitled “Elucidating the role

of alpha synuclein (SNCA) and cholesterol on regulating the P2RX7 pore dilation”

No.F.740/Biochem./2022-23/JD(R). We thank the Director, ICAR-IVRI and Joint Director

Research ICAR-IVRI for their kind support. We thank the head, Division of Biochemistry for his

valuable support and cooperation to execute the research project. We would like to thank the

43

faculty of Biochemistry, Dr. Praveen Singh, Dr. Ajay Kumar and Dr. Meeta Saxena for their

valuable guidance and useful feedback. We thanked technical officer Puja and attendant Riyasat Ji

for his valuable support during conducting research in the lab. We would like to thank Dr.

Himanshu Khandelia, Asso. Professor, South Denmark University, Denmark for his valuable

suggestions during MD data analysis.”

“Mukesh Kumar

No.F.7-40/Biochem./2022-23/JD(R))

ICAR-Indian Veterinary Research Institute

https://www.ivri.nic.in/

No”

7. We note that you have included the phrase “data not shown” in your manuscript. Unfortunately, this does not meet our data sharing requirements. PLOS does not permit references to inaccessible data. We require that authors provide all relevant data within the paper, Supporting Information files, or in an acceptable, public repository. Please add a citation to support this phrase or upload the data that corresponds with these findings to a stable repository (such as Figshare or Dryad) and provide and URLs, DOIs, or accession numbers that may be used to access these data. Or, if the data are not a core part of the research being presented in your study, we ask that you remove the phrase that refers to these data.

Additional Editor Comments:

The submission has an ideal scientific significance, however, needs to be revised qualitatively to meet the standard for publication in this journal.

Reviewers' comments:

Reviewer's Responses to Questions

**Comments to the Author**

1. Is the manuscript technically sound, and do the data support the conclusions?

Reviewer #1: Yes

Reviewer #2: Yes

Reviewer #3: Yes

2. Has the statistical analysis been performed appropriately and rigorously? 

Reviewer #1: I Don't Know

Reviewer #2: Yes

Reviewer #3: Yes

3. Have the authors made all data underlying the findings in their manuscript fully available?

Reviewer #1: No

Reviewer #2: Yes

Reviewer #3: Yes

4. Is the manuscript presented in an intelligible fashion and written in standard English?

Reviewer #1: Yes

Reviewer #2: Yes

Reviewer #3: Yes

5. Review Comments to the Author

Reviewer #1: The manuscript is solid, and the hypothesis is well-defined and supported by preliminary evidence. However, some underlying evidence obtained by the author needed to be provided in the manuscript, which negates the journal guideline. For instance, variation in the RMSD of ⍺Syn was not shown. Additionally, data on the spatiotemporal evolution of each protein was not given.

Reviewer #2: The article is a very good one which will open the way in managing neurodegenerative diseases in the future. the author was able to clearly explain the molecular interaction between P2RX7 and Syn in terms of structural pore intracellularly and extracellularly.

However, the article would be much better by having table of content. also, the method section ----- structure analysis should be change to structural analysis.

Reviewer #3: This is a good manuscript that is well written, However, the authors should endeavor to correct the little errors identified and highlighted below.

- The title is too generic, and may discourage readers, as this will be mistaken for another docking and attractive study, the problem the research addressed should be factor into the Title to make it more attractive and appealing.

- Spacing in between words and numbers such as which84, and24 spaced as “which 84”, as seen in 19

6. PLOS authors have the option to publish the peer review history of their article (what does this mean? ). If published, this will include your full peer review and any attached files.

**Do you want your identity to be public for this peer review?** For information about this choice, including consent withdrawal, please see our Privacy Policy .

Reviewer #1: No

Reviewer #2: No

Reviewer #3: **Yes: ** Sulyman Olalekan Ibrahim

---

## [Author Response · Author response to Decision Letter 1]

13 Jan 2025

Reviewer-I Response

1) We have addressed the reviewer’s suggestions and made the necessary corrections following the journal's guidelines. The reviewer’s concern regarding the RMSD of ⍺Syn in individual complexes is addressed in the supplementary figure S2A. This figure also allows for an analysis of ⍺Syn, P2RX7, and P2RX7-SNCA complexes from the perspective of time and spatial evolution. Spatio-temporal evolution of the molecular system can be explained by RMSD, RMSF, Radius of gyration, secondary structure analysis and PCA. Already we have shown RMSD (Fig 3A), RMSF (Figure S2C), PCA (Fig 3C & S2E)) and Secondary structure analysis (Fig S4) of each protein which can indicate the spatio-temporal evolution of conformational space. However, apart from that we have again added RMSD (Fig S2A), Radius of gyration (Rg) (Fig S2B), Conformational dynamics (PC1 vs PC2 and PC1 vs PC3 projection in 2D graphs) of individual trajectories for 200 ns timescales (Fig S2F) in order to show the time and space evolutions of conformational space of individual P2RX7-SNCA complexes. The figures are shown in Supplementary material and Response to reviewer letter.

2)Review-1 comments within manuscript (PONE-D-24-40709) contents: All the highlighted concerns and suggestions shown in the manuscript contents by reviewer-1 are rectified, supplemented with respective information and shown as track change and markup.

Reviewer-2 response:

Response: We have created the Table of contents but could not find the right place in the manuscript to keep it (table is in Response to reviewer letter). The structure analysis in the method section is replaced by “Structural analysis”.

Reviewer-3 response:

Response: The title of the research article is changed into “Mechanistic insights into Alpha-Synuclein binding to P2RX7: A molecular dynamic and docking Study”.

Response: The typographical mistake was resolved.

---

## [Decision Letter · Decision Letter 1]

28 Jan 2025

Mechanistic insights into Alpha-Synuclein binding to P2RX7: A molecular dynamics and docking Study

PONE-D-24-40709R1

Dear Dr. Kumar,

We’re pleased to inform you that your manuscript has been judged scientifically suitable for publication and will be formally accepted for publication once it meets all outstanding technical requirements.

Kind regards,

Yusuf Oloruntoyin Ayipo, Ph.D

Academic Editor

PLOS ONE

Additional Editor Comments (optional):

The authors have responded appropriately to all my initial concerns and those of the reviewers. They have equally implemented the suggestions for improving the quality of the submission to the standard for publication in this journal. I therefore recommend its publication in the current form.

Reviewers' comments:

Reviewer's Responses to Questions

**Comments to the Author**

1. If the authors have adequately addressed your comments raised in a previous round of review and you feel that this manuscript is now acceptable for publication, you may indicate that here to bypass the “Comments to the Author” section, enter your conflict of interest statement in the “Confidential to Editor” section, and submit your "Accept" recommendation.

Reviewer #1: All comments have been addressed

Reviewer #3: All comments have been addressed

2. Is the manuscript technically sound, and do the data support the conclusions?

Reviewer #1: Yes

Reviewer #3: Yes

3. Has the statistical analysis been performed appropriately and rigorously? 

Reviewer #1: Yes

Reviewer #3: Yes

4. Have the authors made all data underlying the findings in their manuscript fully available?

Reviewer #1: Yes

Reviewer #3: Yes

5. Is the manuscript presented in an intelligible fashion and written in standard English?

Reviewer #1: Yes

Reviewer #3: Yes

6. Review Comments to the Author

Reviewer #1: (No Response)

Reviewer #3: All recommended corrections suggested to the authors have been effected, I therefore recommending acceptance of manuscript.

7. PLOS authors have the option to publish the peer review history of their article (what does this mean? ). If published, this will include your full peer review and any attached files.

**Do you want your identity to be public for this peer review?** For information about this choice, including consent withdrawal, please see our Privacy Policy .

Reviewer #1: **Yes: ** FAIZAH ALABI

Reviewer #3: No

---

## [Editor Report · Acceptance letter]

PONE-D-24-40709R1

PLOS ONE

Dear Dr. Kumar,

I'm pleased to inform you that your manuscript has been deemed suitable for publication in PLOS ONE. Congratulations! Your manuscript is now being handed over to our production team.

Kind regards,

on behalf of

Dr. Yusuf Oloruntoyin Ayipo

Academic Editor

PLOS ONE